# Deciphering anthropogenic and biogenic contributions to selected NMVOC emissions in an urban area

Arianna Peron[1], Martin Graus[1, 2], Marcus Striednig[1], Christian Lamprecht[1], Georg Wohlfahrt[3], Thomas Karl[1]

[1]Department of Atmospheric and Cryospheric Sciences, University of Innsbruck, Innrain 52f, 6020 Innsbruck, Austria
[2] now at IONICON Analytic, Eduard Bodem Gasse, 3, 6020 Innsbruck, Austria
[3]Department of Ecology, University of Innsbruck, Sternwartestrasse 15, 6020 Innsbruck, Austria

*Correspondence to*: Thomas Karl (thomas.karl@uibk.ac.at) and Arianna Peron (arianna.peron@uibk.ac.at)

**Abstract.** The anthropogenic and biogenic contributions of isoprene, monoterpenes, sesquiterpenes and methanol in an urban area were estimated based on direct eddy covariance flux observations during four campaigns between 2018 and 2021. While these compounds are typically thought to be dominated by biogenic sources on regional and global scales, the role of potentially significant anthropogenic emissions in urban areas has been recently debated. Typical fluxes of isoprene, monoterpenes and sesquiterpenes were on the order of $0.07^{+0.02}_{-0.02}$ nmol m$^{-2}$ s$^{-1}$, 0.09 nmol m$^{-2}$ s$^{-1}$ and 0.003 nmol m$^{-2}$ s$^{-1}$ during spring. During summer, emission fluxes of isoprene, monoterpenes and sesquiterpenes were higher on the order of $0.85^{+0.09}_{-0.09}$ nmol m$^{-2}$ s$^{-1}$, 0.11 nmol m$^{-2}$ s$^{-1}$, 0.004 nmol m$^{-2}$ s$^{-1}$. It was found that the contribution of the anthropogenic part is strongly seasonally dependent. For isoprene the anthropogenic fraction can be as high as 64 % in spring, but is typically very low < 18 % during the summer season. For monoterpenes the anthropogenic fraction was estimated between 43 % in spring and less than 20 % in summer.

With values of 2.8 nmol m$^{-2}$ s$^{-1}$ in spring and 3.2 nmol m$^{-2}$ s$^{-1}$ in summer, methanol did not exhibit a significant seasonal variation of observed surface fluxes. However, there was a difference in emissions between weekdays and weekends (about 2.3 times higher on weekdays in spring). This suggests that methanol emissions are likely influenced by anthropogenic activities during all seasons.

## 1 Introduction

Volatile organic compounds (VOCs) are omnipresent in the troposphere due to biogenic or anthropogenic activities (Guenther et al., 1995; Kansal, 2009; Sahu, 2012), and the VOC composition in urban environments can be particularly complex (Calfapietra et al., 2013; Khare et al., 2020; Pfannerstill et al., 2023). Besides substantial anthropogenic VOC (AVOC) emissions, urban environments are home to vegetation and thus also experience the emissions of biogenic VOC (BVOC) (Bonn et al., 2018; Churkina et al., 2017; Ren et al., 2017; Simon et al., 2019). An example of AVOCs are BTEX (benzene, toluene, ethylbenzene and xylenes), which are among the best-studied groups of AVOCs due to their relatively high prevalence in the

urban atmosphere and their potential health risks (Dehghani et al., 2018). BTEX has been linked to combustion and evaporative emissions from industry and transport (Zalel et al., 2008).

The presence of BVOCs in the atmosphere must also be taken into account due to the complexity of the urban environment. However, the characterization of the magnitude of emissions of BVOC in urban areas is still an issue that is being investigated, and it was found that particularly the partitioning of emission sources is complex and often difficult to interpret (Li et al., 2018; Watson et al., 2001; Wei et al., 2008; Wei et al., 2011).

While AVOCs are regionally important, on global scale BVOCs are estimated to be emitted in 10 times higher amounts than
AVOCs (Piccot et al., 1992; Guenther et al., 2012). Of these biogenic emissions, terpenes dominate, of which 50 % can be attributed to isoprene (Guenther et al., 2012). Other terpenes important due to their reactivity in the atmosphere are monoterpenes, accounting for 15 % of total emissions, and sesquiterpenes, with annual emissions of 0.5 % (Guenther et al., 2012). After isoprene, the second largest BVOC emitted by the vegetation is methanol, which can play an important role in global atmospheric chemistry (Tie et al., 2003). The emission of methanol from biogenic source is related to plant growth that
accounts for 40-80 % of the total emission strengths of methanol (Singh et al., 2000; Galbally and Kirstine, 2002; Jacob et al., 2005; Wohlfahrt et al., 2015).

The presence of VOCs , both AVOCs and BVOCs, can affect air quality by forming tropospheric ozone (Derwent et al., 1996) and leads to the formation of precursors necessary for the production of secondary organic aerosols (SOA) (Goldstein et al., 2009; Laothawornkitkul et al., 2009; Riipinen et al., 2012; Simon et al., 2020). In particular, an increasing number of studies
report on the role of isoprene in the formation of SOA. Among these studies, we can mention that of Wu et al. (2020), the result of this study showed how the formation of biogenic SOA depends on the interactions between anthropogenic and biogenic BVOCs emissions. It also pointed out that management of anthropogenic emissions would lead to a reduction in SOA of both biogenic and anthropogenic origin. It is therefore important to understand whether there are variations in the emissions of these compounds, particularly in terms of what measures can be taken to improve air quality in urban environments.

Considerations of BVOC emission management are also reported in Gu et al. (2021) and Pfannerstill et al. (2023) for the city of Los Angeles. These studies analyse the influence of urban greening in emission management on SOA and ozone formation, and how biogenic inventories underestimate isoprene fluxes, with a focus on analysing these fluxes to determine ozone and SOA formation potentials.

Several studies have shown significant interannual variability in BVOC emissions over the years (Palmer et al., 2006; Vaughan
et al., 2017; Kaser et al., 2022). Some factors, such as heatwaves and/or drought, can lead to an increase of the emission of isoprene and monoterpenes by vegetation (Warneke et al., 2010). Mechanical damage of plants within the frame of urban gardening operations has also been observed to play a role for the emission of terpenes (Kaser et al., 2013). Studies using models to determine the influence of future temperature changes on emissions (Steinbrecher et al., 2009; Tawfik et al., 2012) have suggested increased terpenoid fluxes.

To some extent increased emissions can be associated with protective systems against abiotic stressors (Peñuelas and Munné-Bosch, 2005; Velikova et al., 2005). In view of the fact that future climate scenarios predict an increase in temperatures (IPCC,

2007), a possible increase in BVOC emissions worldwide in the range of 30 to 45 % with an increase of 2-3 °C in mean global temperature has been inferred (Peñuelas and Llusiá, 2003). On the other hand, it has been estimated that as $CO_2$ emissions increase, there will be an inhibition of terpenoid emissions (Holopainen et al., 2018, Rosenstiel et al., 2003). From a general point of view, therefore, the estimation of BVOC emissions considering various parameters (IPCC, 2021) is complex, as the dependence of these emissions is related to the evolution of climate and land use scenarios (IPCC, 2021).This has important ramifications for air quality, because terpene emissions play a fundamental role in the chemical reactions that form ozone and secondary organic aerosol (SOA) (Llusiá et al., 2002; Wu et al., 2020; Salvador et al., 2020).

Many studies (Dal Maso et al., 2005; Hellén et al., 2006; Aaltonen et al., 2011; Mäki et al., 2017) have identified the summer period as the time when terpenoids emissions are predominantly of biogenic origin. This time of the year is linked to the growing season, when leaf temperatures and leaf densities are highest (Spirig et al., 2005; Laffineur et al., 2011; Acton et al., 2016). On the other hand, the source of emissions during the late winter to spring period has been investigated less frequently, especially in an urban context.

Due to the low vegetation cover and the dormancy period of plants, the emission of terpenoids during this period of the year was considered to be more related to anthropogenic sources (Hellén et al., 2012; Rouvière et al., 2006). More recently, a number of studies (e.g. McDonald et al., 2018; Gkatzelis et al., 2021a; Coggon et al., 2021) have suggested that urban volatile chemical product (VCP) emissions are a major source of urban terpenoid compounds. This once again highlights the importance of distinguishing between the different sources of emissions of these compounds in terms of their impact on air quality (Chiemchaisri et al., 2001; Claeys et al., 2004a, 2004b; Curci et al., 2010). Indeed, the winter and spring seasons, with their low temperatures, are typically associated with higher emissions of compounds from residential heating, and to some extent also traffic (Shindell et al., 2011; Saha et al., 2018; Squires et al., 2020). Borbon et al. (2023) report that traffic-related monoterpene emissions may account for around 40 % of the environmental levels in the developing world's cities. While on a global scale volatile terpenoids are thought to be mostly emitted from biogenic sources (Guenther et al., 2012), impacting air quality (Chiemchaisri et al., 2001; Claeys et al., 2004; Curci et al., 2010), local urban emissions can also exhibit an anthropogenic component (McDonald et al. 2018). It has been shown that terpenes can be regarded as an important subclass of volatile chemical product (VCP) use in urbanized areas (McDonald et al., 2018; Gkatzelis et al., 2021a; Coggon et al., 2021).

The main aim of this study was thus to investigate the extent of possible anthropogenic contributions to the total emissions of terpenoids (defined as isoprene, monoterpenes and sesquiterpenes) and methanol during two key seasons ((i) late winter – early spring and (ii) summer), by attributing potential biogenic and anthropogenic emissions to changing factors of urban emission activities.

To investigate urban VCP and BVOC emissions we provide a synthesis of four campaigns when direct eddy covariance flux observations were conducted in 2018, 2020 and 2021. These campaigns span different seasons from late winter, early spring to summer, which allows to constrain seasonal urban emission sources of these important VOCs, and estimate the anthropogenic ("VCP") share of isoprene, monoterpenes, sesquiterpenes and methanol emissions in the city of Innsbruck.

## 2 Material and methods

### 2.1 Field site and instruments

The data used in this study originates from four campaigns that took place in 2018 (29 March to 16 April, and 28 July to 31 August), in 2020 (18 March to 12 April) which coincides with the first lockdown (16 March to 1 May 2020) due to the emergence of the SARS-CoV-2 virus (Austrian COVID-19-legislation (BGBl. II Nr. 98/2020)), and in 2021 (25 February to 12 April). The spring trends of VOCs during these three years are also compared to those analysed previously, at the same site, by Kaser et al. (2022).

VOC eddy covariance measurements were conducted at the Innsbruck Atmospheric Observatory (IAO) (47°5′51.66″ N, 11°23′06.82″ E). The measurements were made from a tower at the top of the university building. The tower is 42 m above street level and 13.3 m above the zero-plane displacement height. Estimated at 70 % of the average building height of 19m, this corresponds to the five to seven storey buildings that are most important in terms of drag (Ward et al., 2022). The average building height within 500 m of the IAO is 17.3 m, reduced by the small buildings in the courtyards, which do not have an impact on the flow (Christen et al., 2009). The roughness-length, z0, is 1.6 m (Ward et al., 2022). The dominant wind direction at the IAO is from the NE during the day and SW during nighttime (Karl et al., 2020; Striednig et al., 2020). A climatological analysis of the flux footprint for the different campaigns is presented in the SI. Briefly, during all intensive operational periods (IOPs) most of the footprint (<80%) was contained within a domain extending about 1km to the East and 1 km to the West. During the 2018 spring IOP, we see a slightly higher influence of Föhn conditions (to a lesser extent also 2019), that increases the footprint contribution from the south sectors, compared to spring 2020 and 2021. Overall, the area of interest is quite comparable during all IOPs.

Meteorological data such as temperature, wind speed and direction, and $CO_2$ and $H_2O$ concentrations were obtained using the CPEC200 (Campbell Scientific, USA) at a sampling frequency of 10 Hz. Photosynthetically active radiation (PAR) was estimated based on short wave radiation measurements. A four-component radiometer (CNR4, Kipp & Zonen) at a height of 42.8 m provided incoming and outgoing shortwave and longwave radiation. Air temperature and relative humidity are measured by the HC2S3 probe from Campbell Scientific. Precipitation was measured by a tipping bucket rain gauge (ARG100, Campbell Scientific).

For the interpretation of VOC fluxes in relation to anthropogenic emissions, we analysed $NO_x$ and $O_3$ fluxes. Nitrogen oxides were measured by chemiluminescence (CLD899, Ecophysics, Switzerland), and ozone was measured by absorption spectroscopy (APOA-360, Horiba, Jp) (Lamprecht et al., 2021).

Two proton transfer reaction time-of-flight mass spectrometers (PTR-ToF-MS) were used to obtain BVOC flux data for the campaigns. A proton transfer reaction quadrupole interface time-of-flight mass spectrometer (PTR-QiToF-MS, IONICON Analytik, Sulzer et al., 2014) was used for the 2018 campaigns. A PTR-ToF 6000X2 (IONICON Analytik GmbH, Innsbruck, Austria; Barber et al., 2012; Sulzer et al., 2014) was used for the 2020 and 2021 campaigns. All campaigns employed the same

standard calibration gas (Peron et al., 2021). Kaser et al. (2022) and Peron et al. (2021) describe the operational characteristics

of the instrument for the 2018 and 2020 and 2021 campaigns, respectively. The PTR-ToF-MS was operated at an E/N of about 108 Td to minimize fragmentation. Both PTR-ToF-MS instruments have sufficient mass resolution and mass accuracy to obtain isobaric formulae (Graus et al. 2010). This minimizes potential interferences compared to quadrupole mass spectrometers. The PTR-ToF-MS was used to detect isoprene, at m∕z 69.070 [$(C_5H_8)H^+$], the sum of the monoterpenes at m∕z 137. 133 [$(C_{10}H_{16})H^+$], the major fragment at m∕z 81.070 [$(C_6H_8)H^+$] and the sum of sesquiterpenes at m∕z 205.195 [$(C_{15}H_{24})H^+$]

(Brilli et al., 2011, 2016; Tasin et al., 2012; Maja et al., 2014; Misztal et al., 2015; Giacomuzzi et al., 2016; Yener et al., 2016; Portillo-Estrada et al., 2017).

Recently an evaluation of PTR techniques showed varying fragmentation patterns in urban environments (Coggon et al., 2023). Similarly, for m/z 69.070 Fall et al. (2001) already reported potential fragmentation from $C_5$ alcohols and aldehydes, which would represent an interference for isoprene. Based on earlier evidence (Fall et al., 2001), the most common fragmenting

species are often associated with m/z 87.081. From a regression analysis between m/z 69.070 to m/z 87.081 we obtained an upper limit of about 30 % of an interference for isoprene during the spring campaigns (slope 0.16) and 3 % for the summer campaign (slope 0.01). This is based on the assumption that all ions at m/z 87.081 would fragment at m/z 69.070 in a similar way to common aldehydes measured at this mass (Fall et al. 2001). This is likely an upper limit, because it assumes that all of the ions on m/z 87.081 would represent protonated oxygenated VOC fragmenting on m/z 69.070 by loss of $H_2O$. A potential

interference for the springtime analysis is within the range reported by Coggon et al. (2023) (e.g. up to 40 %), but likely <30 % for this study. The same analysis was carried out for protonated $C_8$ carbonyls (m/z 129.128), together with protonated $C_9$ carbonyls  (m/z 143.144) and the parent protonated ion and the protonated dehydration products (m/z 111.117, and m/z 125.133, respectively), obtaining similar results to those obtained for m/z 87. To account for these uncertainties we have provided a likely range for assigning isoprene to m/z 69 by including upper and lower bounds.

In addition to the fluxes of isoprene, monoterpenes and sesquiterpenes, the fluxes of benzene ([$(C_6H_6)H^+$], m/z 79.054), toluene ([$(C_6H_5CH_3)H^+$], m/z 93.070) and $C_4$ alkenes ([$(C_4H_8)H^+$], m/z 57.070, e.g. 2-butene) were analysed, because the emissions of these compounds are strongly linked to anthropogenic activities (Derwent et al., 2000; Dollard et al., 2007). Emissions of furan ([$(C_4H_4O)H^+$], m/z 69.033) was analysed, because it was shown to be a good tracer for periods influenced by biomass burning (Müller et al., 2016; Koss et al., 2018). Methanol was observed on m/z 33.034 ([$(CH_3OH)H^+$]). Methanol is the most abundant

tropospheric non-methane organic gas (Dominutti et al., 2023), which plays an important role in the chemistry of the remote atmosphere (Duncan et al., 2007; Tie et al., 2007).

An automated valve system allowed sequential analysis of ambient air, calibration and instrument background. Every five and a half hours, calibration and background were performed for two minutes each.

**2.2 Data analysis**

The PTR-ToF-MS data were analysed using the IONICON DATA ANALYZER v4 software (Müller et al., 2013) in order to obtain peak integrated stick spectra in ncps for all VOCs of interest. These data were then converted to ppbv based on a calibration standard (Apel Riemer Environmental Inc., Broomfield, CO, USA), containing 15 compounds (Peron et al., 2021). Fluxes were subsequently calculated by the InnFLUX routine (Striednig et al., 2020). This open-source software allows to process eddy covariance and disjunct eddy covariance flux data. The high frequency loss was inferred to amount to 1.1 %

based on spectral analysis (Striednig et al., 2020). Fluxes were filtered by common quality criteria (Foken et al., 2012), which are a combination of the steady-state and integral turbulence characteristic tests. The quality assurance procedures were described previously in Kaser et al. (2022).

To make the interpretation of results consistent with the analysis of Kaser et al., (2022) data were additionally filtered by

analysing fluxes originating from the east sector between 0 and 120° (NE), the dominant wind direction.

Given that it is well established that isoprene emissions are light dependent (Kesselmeier and Staudt, 1999; Guenther et al., 2006), nighttime emissions (between 8 p.m. and 3 a.m. UTC) were assumed to be exclusively of anthropogenic origin, while daytime emissions (between 8 a.m. and 4 p.m.) represent a mix of biogenic and anthropogenic sources.

Significant interference from potentially stressed plants were investigated by correlating with green leave volatiles (GLV).

The emission of GLVs by plants has been attributed to both abiotic and biotic stresses (Halitschke et al., 2004; Loreto et al., 2006; Wenda-Piesk, 2011; Allmann et al., 2013). Compounds belonging to this group, which includes molecules such as esters, alcohols and aldehydes, are characterised by a chain of six carbon atoms ($C_6$). Specifically, for this analysis Hexenal ($[(C_6H_{10}O)H^+]$, m/z 99.080) was considered (Beauchamp et al., 2005; Giacomuzzi et al., 2016; Portillo-Estrada et al., 2017) as a marker compound. The analysis showed no significant correlation due to low fluxes on ions associated with GLV. We

also analysed the fluxes of Decamethylcyclopentasiloxane ($C_{10}H_{30}O_5Si_5$, D5), which is related to the usage of cosmetics and other health care products, allowing to use this compound as a human mobility proxy marker (Horii and Kannan, 2008; Wang et al., 2009).

Regarding the data treatment of monoterpenes and sesquiterpenes, the two categories were analysed using different means to calculate ambient concentrations due to common fragmentation patterns. For monoterpenes, the correlation between exact m/z

137.133 and the exact m/z 81.070 was used to calculate the overall abundance. In all four campaigns, the correlation between parent ions and expected fragments was strong, both for fluxes and for concentrations. From the regression, the concentrations and fluxes of monoterpenes were based on the fragment correct signal on m/z mass 137.133.

Regarding the quantification of sesquiterpenes, common fragments were used to correct for the abundance of sesquiterpenes on m/z 205.195 as by Kaser et al. (2022). Due to the potential of a higher contribution of sesquiterpenes that react very fast

with respect to ozone (e.g. beta- caryophellene) we additionally estimated potential chemical losses due to reaction with ozone as suggested by Kaser et al., (2022). Since the PTR-ToF-MS does not allow to speciate sesquiterpenes, upper and lower limits

for the composition were chosen based on typical literature values. In particular, the sesquiterpene composition mix used by Sakulyanontvittaya et al. (2008) was applied, with a reactive terpene fraction between 36 % and 50 %. Due to the fast reaction with ozone potential losses between the point of emission and detection was estimated. Ozone data were obtained from ambient measurements using a UV absorption technique. Based on the commonly reported composition of sesquiterpenes (Sakulyanontvittaya et al. 2008) we used these constraints, to estimate the possible breakdown of sesquiterpenes fluxes due to chemical reactions in the atmosphere. We obtained a range of approximately 30-45 %.

In order to constrain the influence of the anthropogenic component on the emissions, we investigated the difference in emissions between the weekend (Sunday and public holidays) and weekdays (i.e. working days from Monday to Thursday). The weekend to weekdays effect is defined as the average flux (or activity) difference by defining the ratio of average emissions between weekdays and weekends. This differentiation of the fluxes during two different periods of the week (weekend and weekdays) has also allowed us to estimate the percentage of the anthropogenic or biogenic contribution to the total of the measured emissions. The analysis was done by pooling the data from a narrow temperature window of (T = 278 - 282 K). This consideration has been made to minimize the effect of a potential temperature dependence of the BVOC emissions. In order to maintain a statistically significant number of data points, it was decided to aggregate all of the data from the three spring campaigns first. To determine whether the weekend to weekdays ratio is statistically significant, a t-test was applied to the data using 50 random realizations.

## 2.3 Emission modelling

BVOC emission factors in MEGAN (Guenther et al., 2012) for urban areas are still scarce (Leung et al., 2010). In addition, during the winter and spring months, AVOC emissions can mask BVOC emissions and the anthropogenic fraction can potentially dominate.

In order to include anthropogenic and biogenic sources for urban isoprene emissions one can partition the dataset between an anthropogenic flux component and a biogenic component parameterized by MEGAN.

For isoprene, both terms can be modelled independently to reconstruct the total emission flux ($F_{VOC}$): according to (Eq.1).

$$F_{VOC} = F_{AVOC} + F_{BVOC} \qquad (1)$$

The variation of the anthropogenic component for isoprene was calculated from nighttime data, when biogenic isoprene emissions can be assumed to be negligible, by correlating isoprene fluxes with benzene fluxes ($F_{Be}$). The urban benzene flux, is thought to be dominated by urban combustion processes in Innsbruck particularly traffic related emissions. The ratio of isoprene to benzene emissions ($\alpha$) over the same period can then be estimated according to (Eq. 2):

$$F_{VOC,ant} = \alpha \times F_{AVOC} \qquad (2)$$

$\alpha$ was calculated based on a linear x-y weighted fit between nocturnal isoprene and benzene fluxes (Cantrell., 2008), assuming that all the isoprene emissions during nighttime originate from anthropogenic sources. Unfortunately, this correlation analysis

cannot be applied to monoterpenes and sesquiterpenes, because it is not reasonable to assume that nighttime biogenic emissions of these compounds are negligible. We therefore resort to an independent analysis based on the weekend to weekday effect for these compounds.

To analyse the correlation with benzene fluxes we used a linear fit in Matlab with uncertainties both in x and y (linfitxy, Browaeys, 2023).

The inferred biogenic isoprene fluxes were subsequently normalized based on common biogenic emission algorithms (e.g. MEGAN, Guenther et al., 2012, eq. 3).

$$F_{BVOC} = E_0 \gamma_L \gamma_T \tag{3}$$

where $\gamma_L$ represents the light dependent activity factor, and $\gamma_T$ the short-term temperature dependent activity factor.

In this study, the stand-alone version of MEGAN (Guenther et al., 2012), was used to calculate the light and temperature variation of isoprene emissions in response to T [C°] and PAR [$\mu$mol m$^{-2}$ s$^{-1}$]. In order to determine the biogenic isoprene emission factor, eq. 3 was used to normalize measured fluxes after subtracting the anthropogenically derived portion of the isoprene flux that was inferred from the nighttime correlation analysis (Eq. 4):

$$E_0 = \frac{(F_{IS} - \alpha F_{Be})}{\gamma_L \gamma_T} \tag{4}$$

## 2.4 Ancillary data

In order to characterize the seasonal vegetation phenological development in the study area, the normalised difference vegetation index (NDVI) was used (Rouse et al. 1974). To this end, the harmonized Landsat-Sentinel HLSS30 v002 satellite product (Masek et al. 2021) was downloaded through NASA's AppEEARS web interface (https://appeears.earthdatacloud.nasa.gov). The HLSS30 product is derived from Sentinel-2A and -2B products and provides Nadir BRDF-corrected reflectances with a 30 m spatial and 2-3 day temporal resolution. The NDVI was calculated from the narrow-band reflectances in the red (band 4) and near-infrared (band 8A). Data were filtered for the presence of surface water, ice and snow, as well as clouds, clouds shadows and cloud shadows in adjacent pixels using the quality assessment layer provided with the product. Data were downloaded for the period 2018-2022 and cut out for a 1 x 1 km area with the flux tower in the centre. Overall, 35 % of the data passed quality control, during the measurement campaigns this fraction ranged between 27 and 47 %. The final data were weighted with the flux footprint calculated using the model by Kljun et al. (2015). To that end, satellite NDVI data were linearly interpolated in time between satellite retrievals to the flux timestamps.

## 3 Results and discussion

### 3.1 Meteorological, urban metabolism and traffic overview

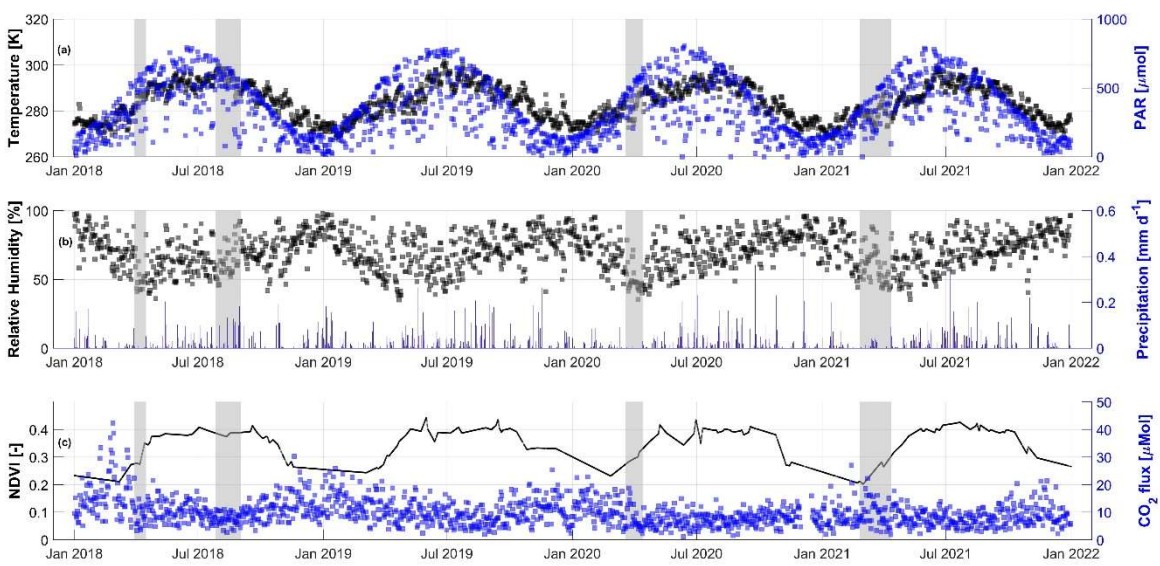

**Figure 1: Daily meteorological conditions and urban metabolism between January 2018 and January 2022. Panel (a) air temperature [K] – left y-axis, PAR [μmol m$^{-2}$s$^{-1}$] – right y-axis. Panel (b) the relative air humidity [%] – left y-axis, daily precipitation [mm d$^{-1}$] – right y-axis. Panel (c) NDVI – left y-axis, and CO$_2$ flux [μmol m$^{-2}$s$^{-1}$] – right y-axis. The grey areas indicate the duration of the measurement campaigns.**

The analysis of the weather conditions over the three years under consideration shows no particular variations in PAR and relative humidity (Fig. 1). In fact, the three spring periods are comparable when analysing these two parameters.

With regard to temperature, the average daytime temperature in spring 2021 was lower (9.60 °C) than in 2018 and 2020 (15.2 °C and 15.10 °C respectively). During the night, on the other hand, a decrease in the average temperature was observed over the three years, from 9.40 °C in 2018 to 2.50 °C in 2021 (Table 1). These variations are related to the different sampling periods, which are not entirely coincident.

Analysing the precipitation during the three spring periods, it can be seen that the spring of 2020 was the one with the lowest precipitation (2.3 mm), although the campaign only lasted 26 days. On the other hand, 2021 is the year in which more precipitation was recorded (56.7 mm) than in 2018 and 2020 (Table 1).

The plant phenological development for the spring campaigns in 2018, 2020, 2021 shows no significant inter-annual differences with average NDVI values of 0.28, 0.28 and 0.25, respectively (Table 1). For the summer season campaign 2018, the average value was 0.37 (Table 1), similar to what was observed during summers 2019-2021 (Fig. 1). Generally, a low vegetation cover characterizes the flux footprint - Ward et al. (2021) showed that on average vegetation cover comprises about 19 % of the flux footprint surrounding the site.

The dominant source of $CO_2$ emissions in Innsbruck are anthropogenic emissions (i.e. traffic and residential heating) (Ward et al., 2022; Nicolini et al. 2022). The seasonal variation (Fig 1.) clearly shows this with higher $CO_2$ emission in the cold season than during the warm season, when $CO_2$ fluxes decrease. Lower $CO_2$ emissions during summer reflect reduced anthropogenic emissions (50 % of $CO_2$ emissions are estimated to originate from heating) and, to a minor extent, higher photosynthetic uptake. Previous work at this site suggested a $CO_2$ flux bias of about 10-20 % during summer corresponding to the vegetated land cover surrounding the flux tower (Ward et al., 2022).

**Table 1: Average daytime and nighttime temperature [°C], PAR [$\mu$mol m$^{-2}$s$^{-1}$], relative humidity [%], $CO_2$ flux [$\mu$mol m$^{-2}$s$^{-1}$]. NDVI [-] values are referring to the averaged value of the single period. Sum of precipitation [mm] are the sums of the seasonal precipitation. The data are reported with corresponding standard errors for the four measurement campaigns.**

| | Spring 2018 | | Summer 2018 | | Spring 2020 | | Spring 2021 | |
|---|---|---|---|---|---|---|---|---|
| | Daytime | Nighttime | Daytime | Nighttime | Daytime | Nighttime | Daytime | Nighttime |
| Temperature [°C] | 15.2 ± 0.08 | 9.40 ± 0.07 | 27.20 ± 0 08 | 18.40 ± 0.06 | 15.10 ± 0.10 | 5.80 ± 0.08 | 9.60 ± 0.07 | 2.50 ± 0.05 |
| PAR [$\mu$mol m$^{-2}$s$^{-1}$] | 965.25 ± 11.28 | - | 1228.50 ± 8.83 | - | 1257.00 ± 8.33 | - | 839.25 ± 6.14 | - |
| Relative Humidity [%] | 37.00 ± 0.35 | 63.00 ± 0.33 | 41.00 ± 0.28 | 78.00 ± 0.21 | 30.00 ± 0.24 | 62.00 ± 0.24 | 41.00 ± 0.24 | 73.00 ± 0.19 |
| $CO_2$ [$\mu$mol m$^{-2}$s$^{-1}$] | 4.70 ± 0.24 | 9.09 ± 0.33 | 2.88 ± 0.12 | 9.42 ± 0.20 | 4.74 ± 0.17 | 8.14 ± 0.21 | 4.26 ± 0.16 | 9.93 ± 0.21 |
| NDVI [-] | 0.2772 ± 0.0013 | | 0.3838 ± 0.0001 | | 0.2935 ± 0.0005 | | 0.2557 ± 0.0007 | |
| Sum of precipitation [mm] | 25.4 | | 137.0 | | 2.3 | | 56.7 | |

With regard to anthropogenic activities, the effect from weekend to weekdays was analysed. Results are plotted in Figure 2, which shows that this ratio varies depending on the type of compound and/or activity (Fig. 2). To determine whether the weekend to weekday ratio is statistically significant, a t-test was applied to the data using 50 random realizations This statistical analysis shows that when the three spring campaigns are analysed together, there is a significant difference between the weekday and weekend fluxes for isoprene, monoterpenes, benzene, toluene, methanol, $C_4$ alkenes, $NO_x$ and traffic activity. No significant difference was found for sesquiterpenes, siloxanes, temperature and PAR.

For compounds such as isoprene, monoterpene, benzene, toluene-methanol, $C_4$ alkenes (e.g. 2-butene) and $NO_x$, emissions are highest during the week, i.e. Monday to Thursday. This seems to indicate that the emission of these compounds is largely driven by anthropogenic activities. In the joint analysis of the three spring campaigns, compounds such as sesquiterpenes and siloxanes do not seem to be influenced by a change in anthropogenic activities (Fig. 2), as the average flux values do not show any particular variations between the two identified weekly periods. In the case of siloxanes, this would seem to indicate that the population maintains the same personal care habits during weekdays as during the weekend.

The traffic volume was found to be on average about 3.8 times higher on weekdays than on weekends. In addition, and in order to better understand the possible influence on the emissions of the VOC of interest considered in this study, an analysis of traffic flow (transit of motorized vehicles) and $NO_x$ fluxes was carried out (Fig. 2). Traffic, $NO_x$, benzene and toluene emissions

differ significantly between the weekend and weekday periods. For traffic, it was estimated that the number of vehicles on the road during the weekdays is almost four times higher than at the weekend. Comparable variations were found for $NO_x$, benzene and toluene for which workday emissions are about three times higher than on the weekend.

To exclude the possibility of varying environmental conditions for this type of analysis we compared PAR and air temperature during weekdays and weekends. PAR did not show a statistically significant difference. To check whether there is still a remaining temperature variation after the data selection, the analysis was also performed for air temperature and, as expected, no variation was found. These results strongly suggest that the weekday-weekend differences of all VOCs are most likely related to a variation in anthropogenic activities.

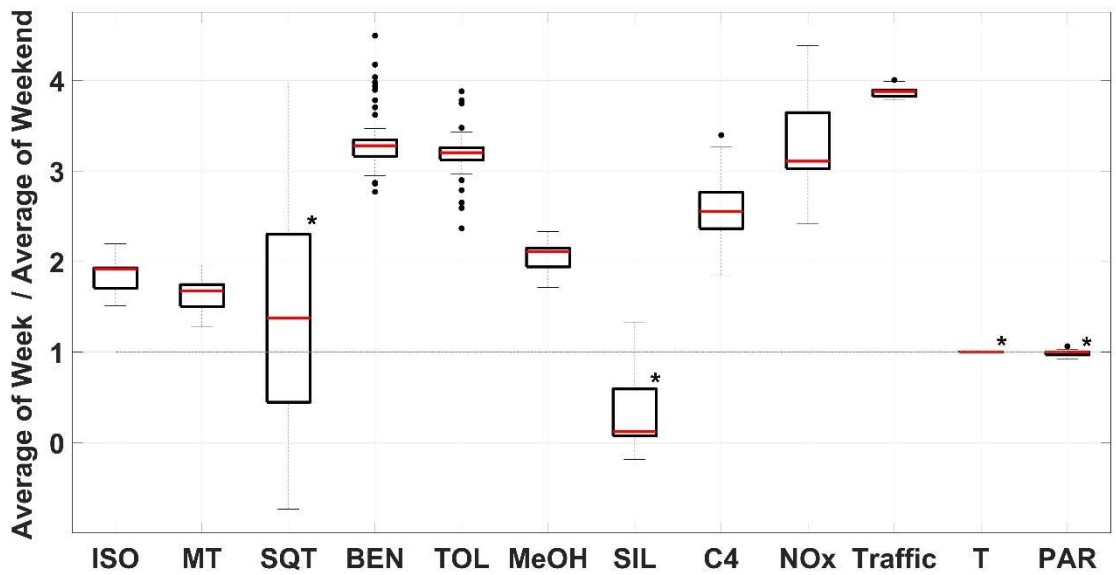


**Figure 2: Data from all campaigns plotted as a ratio between the mean value during the weekdays and the weekend period for isoprene (ISO), monoterpenes (MT), sesquiterpenes (SQT), benzene (BEN), toluene (TOL), methanol (MeOH), siloxane (SIL), $C_4$ alkenes (C4), NOx, traffic, temperature (T) and PAR. * represent cases where the t-test for the weekend effect on the data (p-value > 0.05) was not statistically significant.**

As a next step, the analysis was carried out on a campaign-by-campaign basis, bearing in mind that this analysis is less robust due to the limited number of weekend data for each campaign. In this case, for the summer data, only fluxes emitted when the temperature was higher than 292 K were taken into account in order to minimize the temperature-related effect on BVOC emissions. The analysis for spring time data was the same as for the bulk analysis described above. An analysis of the weekend effect for VOCs during the different campaigns shows that the situation changes depending on whether the period is spring or

summer (Fig. 3). The variation of the compounds will be analysed in the following chapters.

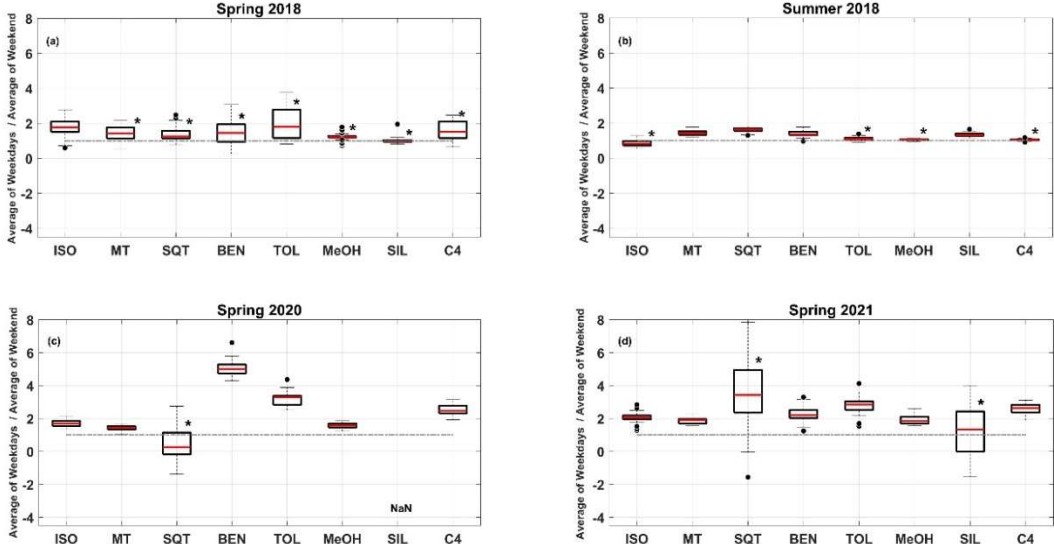

**Figure 3: Ratio between average values during weekend and weekdays of VOCs fluxes for spring 2018 (a), summer 2018 (b), spring 2020 (c) and spring 2021 (d) for fluxes of isoprene (ISO), monoterpenes (MT), sesquiterpenes (SQT), benzene (BEN), toluene (TOL), methanol (MeOH), siloxane (SIL) and C4 alkenes (C4). \* represent cases where the t-test for the weekend effect on the data (p-value > 0.05) was not statistically significant.**

### 3.2 Isoprene

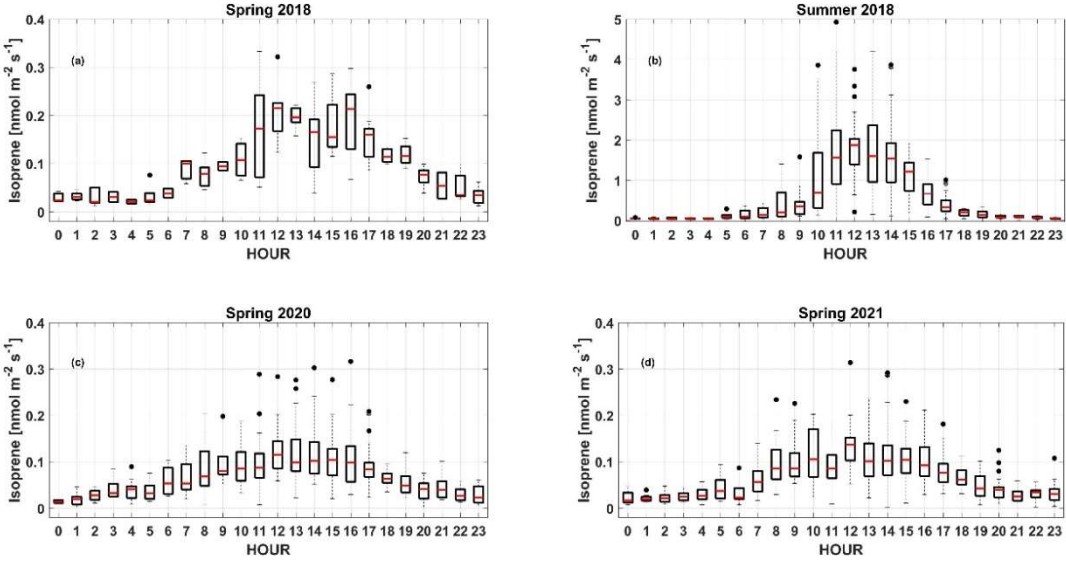

**Figure 4: Hourly averaged isoprene fluxes during spring 2018 (a), summer 2018 (b), spring 2020 (c) and spring 2021 (d). Note the different y-scale in panel (b).**

Figure 4 shows daily variations of isoprene fluxes during all campaigns. For the three spring periods, respectively, average isoprene fluxes amounted to $0.09^{+0.03}_{-0.02}$ nmol m$^{-2}$ s$^{-1}$ for 2018, $0.05^{+0.01}_{-0.01}$ nmol m$^{-2}$ s$^{-1}$ for 2020 and $0.05^{+0.01}_{-0.01}$ nmol m$^{-2}$ s$^{-1}$ for 2021. In spring, on average, 64 % of the measured isoprene flux was attributed to anthropogenic emissions according to equation 3. The modelled average anthropogenic isoprene emissions were $0.05^{+0.01}_{-0.01}$ nmol m$^{-2}$ s$^{-1}$ in the year 2018, $0.03^{+0.01}_{-0.01}$ nmol m$^{-2}$ s$^{-1}$ in the year 2020 and $0.04^{+0.01}_{-0.01}$ nmol m$^{-2}$ s$^{-1}$ in the year 2021.

For comparison, the anthropogenic fraction was quite a bit lower in summer compared with the total amount of the isoprene flux ($0.15^{+0.03}_{-0.03}$ nmol m$^{-2}$ s$^{-1}$, 18 % of total isoprene emissions). This is consistent with findings of Kaser et al. (2022).

    Isoprene can also be released from burning activities (Yokelson et al., 2009). To rule out any potential interference of burning activities during nighttime on isoprene fluxes, furan fluxes, which were proposed to be a good marker for urban biomass burning (BB) phenomena (Koss et al., 2018), were investigated. The averaged nocturnal fluxes of furan were 0.005

nmol m$^{-2}$ s$^{-1}$, 0.009-nmol m$^{-2}$ s$^{-1}$, 0.007-nmol m$^{-2}$ s$^{-1}$, 0.005-nmol m$^{-2}$ s$^{-1}$ in spring 2018, summer 2018, spring 2020 and 2021, respectively. For the corresponding periods, furan fluxes did not exceed 0.01 nmol m$^{-2}$ s$^{-1}$, 0.03 nmol m$^{-2}$ s$^{-1}$, 0.05 nmol m$^{-2}$ s$^{-1}$ and 0.04 nmol m$^{-2}$ s$^{-1}$. These low fluxes suggest that there were no significant BB episodes during the measurement period reported here. This is also supported by the low correlation ($R^2 = 0.39$) between furan and isoprene fluxes and concentrations during the nighttime spring periods. Biomass burning plumes would have clearly influenced the relationship between isoprene

and benzene, due to different emission ratios and correlation (Santos et al., 2018). The data do not show any significant interannual differences for mean fluxes of benzene, neither during the day (0.20 nmol m$^{-2}$ s$^{-1}$ – 2018, 0.10 nmol m$^{-2}$ s$^{-1}$ – 2020, 0.14 nmol m$^{-2}$ s$^{-1}$ – 2021) nor at night (0.04 nmol m$^{-2}$ s$^{-1}$ – 2018, 0.03 nmol m$^{-2}$ s$^{-1}$ – 2020, 0.02 nmol m$^{-2}$ s$^{-1}$ – 2021).

    To test for other potential interferences for nighttime fluxes (i.e. plant stress and damage) for isoprene and terpenoid emissions, we investigated Green Leaf Volatiles (GLV). Emissions of GLV are associated with biotic (Maja et al., 2014; Faiola and

Taipale, 2020) and abiotic stresses (Fall et al., 1999; Beauchamp et al., 2005; Behnke et al., 2009; Betz et al., 2009). In spring during the nighttime period, the mean GLV fluxes were 0.005 nmol m$^{-2}$ s$^{-1}$ in 2018, 0.002-nmol m$^{-2}$ s$^{-1}$ in 2020, and 0.004 nmol m$^{-2}$ s$^{-1}$ in 2021. The maximum values reported were 0.01 nmol m$^{-2}$ s$^{-1}$, 0.03-nmol m$^{-2}$ s$^{-1}$, 0.02 nmol m$^{-2}$ s$^{-1}$ and 0.03 nmol m$^{-2}$ s$^{-1}$ respectively. In summer, the mean GLV flux during the night was 0.005 nmol m$^{-2}$ s$^{-1}$ with a maximum not exceeding 0.03-nmol m$^{-2}$ s$^{-1}$. These low fluxes and little correlation ($R^2 = 0.35$) between GLV and VOC fluxes suggests little

influence due to interfering species from plant stress.

    The diurnal trend of the isoprene emissions, divided into weekdays and the weekend, showed that most of the emission fluxes occur during daylight. Since anthropogenic emissions are expected to be less influenced by temperature variations, higher isoprene fluxes during summer are due to biogenic emissions.

    However, during spring non-zero nighttime fluxes (Fig.4) of isoprene become more prominent due to the overall low isoprene

flux. The nighttime emissions of isoprene must be a result of the presence of anthropogenic sources (e.g. tail-pipe emissions from cars). If true, there should also be evidence from the weekend/weekday effect. Indeed, it was found that nighttime emission fluxes during weekdays were higher compared to the nighttime emission fluxes on weekends. This is because then these fluxes are more influenced by transport volumes (e.g. traffic activity).

During spring the difference between weekends and holidays, and typical working days of the week was statistically significant, with lower emissions during holidays and weekends. (Table 2). The weekend-weekday difference was consistently observed over three years of measurements (Fig. 3). The same statistics during a summer campaign on the other hand, did not show any significant difference, suggesting that summer emissions of isoprene are dominated by biogenic emissions (Kaser et al., 2022). This information supports the hypothesis, that anthropogenic emissions of isoprene play a more important role, in the urban environment, during spring periods.

**Table 2: Average value of the isoprene, monoterpenes and methanol emissions measured during the four campaigns on Sundays and holidays (weekend) and working days (weekdays) with relative p-value for the significance of the difference between the different days of the week. For isoprene to account for the uncertainties we have provided a likely range for assigning isoprene to m/z 69 by including upper and lower bounds.**

| | | Spring 2018 $[\text{nmol m}^{-2}\,\text{s}^{-1}]$ | Summer 2018 $[\text{nmol m}^{-2}\,\text{s}^{-1}]$ | Spring 2020 $[\text{nmol m}^{-2}\,\text{s}^{-1}]$ | Spring 2021 $[\text{nmol m}^{-2}\,\text{s}^{-1}]$ |
|---|---|---|---|---|---|
| Isoprene | Weekdays | $0.05\ ^{+0.02}_{-0.01}$ | $0.15\ ^{+0.02}_{-0.02}$ | $0.05\ ^{+0.02}_{-0.01}$ | $0.06\ ^{+0.02}_{-0.01}$ |
| | Weekend | $0.03\ ^{+0.01}_{-0.01}$ | $0.18\ ^{+0.02}_{-0.02}$ | $0.03\ ^{+0.01}_{-0.01}$ | $0.03\ ^{+0.01}_{-0.01}$ |
| Monoterpenes | Weekdays | 0.11 | 0.07 | 0.09 | 0.09 |
| | Weekend | 0.07 | 0.05 | 0.06 | 0.05 |
| Sesquiterpenes | Weekdays | 0.003 | 0.002 | 2 E-04 | 0.001 |
| | Weekend | 0.003 | 0.001 | 8 E-04 | 3 E-4 |
| Benzene | Weekdays | 0.09 | 0.14 | 0.12 | 0.15 |
| | Weekend | 0.06 | 0.10 | 0.02 | 0.07 |
| Toluene | Weekdays | 0.15 | 0.16 | 0.21 | 0.26 |
| | Weekend | 0.08 | 0.15 | 0.06 | 0.09 |
| Methanol | Weekdays | 1.22 | 2.51 | 3.25 | 3.74 |
| | Weekend | 1.03 | 2.36 | 1.97 | 2.01 |
| Siloxane | Weekdays | 0.003 | 0.001 | 2 E-4 | 1 E-4 |
| | Weekend | 0.003 | 0.001 | NaN | 2 E-4 |
| $C_4$ Alkenes | Weekdays | 0.16 | 0.30 | 0.21 | 0.28 |
| | Weekend | 0.10 | 0.29 | 0.08 | 0.11 |

The correlation analysis based on nighttime flux analysis in conjunction with an additive emission model (eq. 2-4) allows to apportion isoprene fluxes into an anthropogenic and biogenic component. Thus, isoprene emissions during summer are dominated by biogenic emissions. This is in line with Kaser et al. (2022), who estimated that biogenic emissions account for 80-95 % of the observed isoprene flux during summer. We can also use the weekend to weekday effect as an independent estimate to constrain the anthropogenic vs biogenic component of isoprene emissions. From Table 2 and Figure 3 we get a

weekday to weekend variation of isoprene fluxes that is on the order of 1.8 for spring. There is no statistically significant variation during the summer (i.e. the ratio is close to 0.8). This suggests that about 50 % of isoprene fluxes exhibit an anthropogenically driven source activity for spring but not for summer.

We have previously observed that a large number of oxygenated VOCs can be emitted in urban areas (Karl et al., 2018), and factor analysis revealed distinct emission patterns, where aromatic compounds typically cluster with traffic related signals. The flux ratio of toluene to benzene fluxes exhibited a high correlation ($R^2 = 0.79$) with an average flux ratio of 1.6 ([nmol/m$^2$/s] / [nmol/m$^2$/s]) during spring, and 1.9 ([nmol/m$^2$/s] / [nmol/m$^2$/s]) during summer in this study. This supports previous findings in Europe (Schnitzhofer et al., 2008), that toluene emissions from cars are evaporative and temperature dependent throughout the season. In comparison the correlation between monoterpene and benzene was poorer ($R^2 \sim 0.46$) with a typical flux ratio of 0.16 ([nmol/m$^2$/s] / [nmol/m$^2$/s]) in summer and 0.17 ([nmol/m$^2$/s] / [nmol/m$^2$/s]) in spring.

### 3.3 Monoterpenes

#### 3.3.1 Weekday-Weekend effect

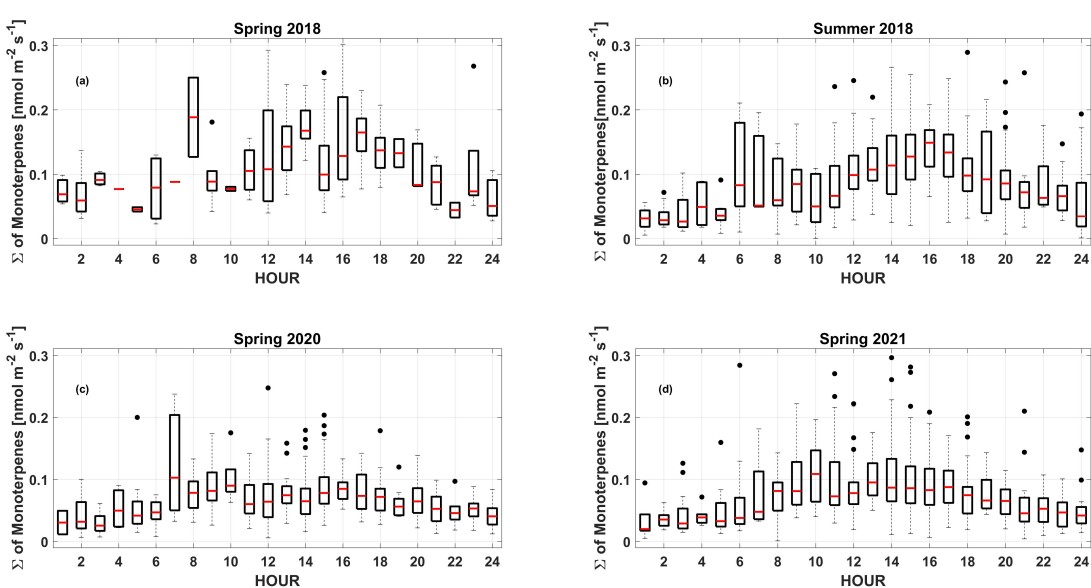

**Figure 5: Daily sum of monoterpenes fluxes during spring 2018 (a), summer 2018 (b), spring 2020 (c) and spring 2021 (d).**

Compared to isoprene, the partitioning analysis of monoterpenes and sesquiterpenes is more complex, because nighttime biogenic emissions may not be assumed negligible a priori. The data show a less pronounced daily flux trend (Fig. 6), as is the case for isoprene. This is because biogenic monoterpene emissions are not solely dependent on the presence of PAR and can have a significant temperature driven emission from storage pools (Guenther et al., 1993, 1995, 2012).

We observed typical emission fluxes of monoterpenes on the order of 0.09 nmol m$^{-2}$ s$^{-1}$ during the daily period in spring 2018, 2020 and 2021, and 1.11 nmol m$^{-2}$ s$^{-1}$ during summer 2018. The higher emissions of monoterpenes in 2018 (0.12 nmol m$^{-2}$ s$^{-1}$) compared to 2020 (0.07 nmol m$^{-2}$ s$^{-1}$) and 2021 (0.08 nmol m$^{-2}$ s$^{-1}$), were not statistically significant (p-value 0.14). The percentage of these emission attributed to biogenic sources was estimated based on the weekend to weekday effect as for isoprene. We see that the weekend to weekday effect for monoterpenes (as for isoprene) was more significant during the spring campaigns 2020 and 2021 (an average ratio of 1.7) than during the summer campaign (1.4). In addition, there also seems to be a decrease in emission fluxes during the weekdays of the spring lockdown period (0.11 nmol m$^{-2}$ s$^{-1}$ - spring 2018, 0.09 nmol m$^{-2}$ s$^{-1}$ - spring 2020, 0.09 nmol m$^{-2}$ s$^{-1}$ - spring 2021). The weekend to weekday ratio in spring 2020 was close to the ratio observed during the 2018 summer campaign. The correlation between benzene and monoterpene fluxes is somewhat poorer than for isoprene (e.g. $R^2 = 0.61$ during the spring campaigns for monoterpenes vs $R^2 = 0.77$ for isoprene (Figure S6 in Supplement)). In spring, the slopes between isoprene/benzene fluxes were 2.1, and for monoterpenes it was 0.9, respectively (data not shown). For summer we find that the correlation coefficient between monoterpenes and benzene was poor, with an $R^2$ of only <0.3 (Figure S7 in Supplement). The anthropogenic fraction of monoterpenes was 43% in spring and 20% in summer. Yet the total monoterpene flux did not change significantly between these seasons like for isoprene. This might come as a surprise, since an increase of biogenic emissions is expected in summer. Our explanation for this is that the increase of biogenic emissions was compensated by a decrease of anthropogenic emissions due to two main factors: (1) The IAO is located close to the city centre at one of the university campuses. In August, most students are on summer break and fewer people are on the streets within the flux footprint. (2) The campaign happened to take place during a significant heat wave in 2018, when people preferentially stayed in cooler environments rather than in the unpleasant street climate of the urban heat island. Both factors provide an explanation that fewer people were out on the streets during the day in the summer of 2018, and that the anthropogenic fraction of monoterpene fluxes was lower. Coggon et al. (2021) reported the ratio between monoterpenes and benzene in correlation with population density, with an $R^2$ of 0.48. They suggested that emissions from drivers and passengers are co-emitted with benzene (emitted from the tailpipe) and therefore exhibit a good correlation in urban areas. To put our results in context of the study of Coggon et al. (2021) we investigated the relationship between monoterpenes and benzene, based on population density. Coggon et al. 2021 obtain a concentration enhancement ratio of monoterpenes / benzene that depends on population density and is on the order of 5.2e-5 [ppb/ppb]/[people/km$^2$] for New York City. From our flux data, we can calculate the ratio of monoterpene to benzene fluxes for the spring IOPs, and normalize it by the population density estimated for the city centre of Innsbruck (7000 – 8800 people/km$^2$) (Ward et al., 2022). We obtain a normalized ratio of 3.9e-5 to 5.0e-5 [(nmol/m$^2$/s)/(nmol/m$^2$/s)]/[people/km$^2$]. Considering that about 43% of springtime monoterpenes fluxes are of anthropogenic emission, we find a normalized flux ratio of 1.7e-5 to 2.2e-5 [(nmol/m$^2$/s)/(nmol/m$^2$/s)]/[people/km$^2$].

### 3.3.2 COVID lockdown period

A special case in this dataset is the period of 2020 during the hard COVID lockdown, when a significant weekday-weekend effect was observed for human markers (Figure 3).

When analysing siloxanes, we observed that D5 fluxes were present during weekdays, but not at weekends. This would seem to indicate that the pedestrian circulation of people was active during the weekday period, due to the need to buy groceries and partially work (which was still permitted). On the other hand, during the weekend period, the pedestrian circulation of people was almost zero, as people were not allowed to leave their homes, except for special reasons. Similar results were observed for monoterpene fluxes.

This is a qualitative indication that, in contrast to isoprene, anthropogenic emissions of monoterpenes are more related to the use of personal care products than due to vehicle traffic (Oz et al., 2015; Nazzaro et al., 2013; Cheng et al., 2018; Zhang et al., 2020; Gkatzelis et al., 2021b; Panopoulou et al., 2020, 2021; Kornbausch et al., 2022).

The correlation between D5 siloxane and benzene fluxes observed here, only exhibited an $R^2$ lower than <0.13, thus was very poor. In contrast, Gkatzelis et al. (2021a) observed a very high correlation with $R^2$ of ~0.8. However, D5 siloxane fluxes observed here are generally very low, especially in spring. From the spring analysis, the weekend to weekday variation of monoterpenes was similar to isoprene (e.g. a ratio of 1.9 for isoprene vs a ratio of 1.6 for monoterpenes). This suggests a similar partitioning between anthropogenic and biogenic emission components in spring. As a best estimate from various constraints, we argue that 50-67 % of isoprene and monoterpenes emissions during spring can be associated with anthropogenic activity, but less than 20 % during summer. In this context it is noteworthy to mention that isoprene fluxes are about a factor of 10 higher during summer (Fig. 4), while monoterpenes fluxes vary much less throughout the seasons and the magnitude is comparable during the seasons (Fig. 5).

In this study, a good correlation between monoterpenes and D5 reported by Gkatzelis et al. (2021a) was not present during the nighttime and during the lockdown period (data not shown). This is an indication that the anthropogenic component of monoterpene fluxes was lower during the lockdown period, and consequently was more associated with vegetation due to the strict mobility restrictions in Austria (Lamprecht et al., 2021).

**3.4 Sesquiterpenes**

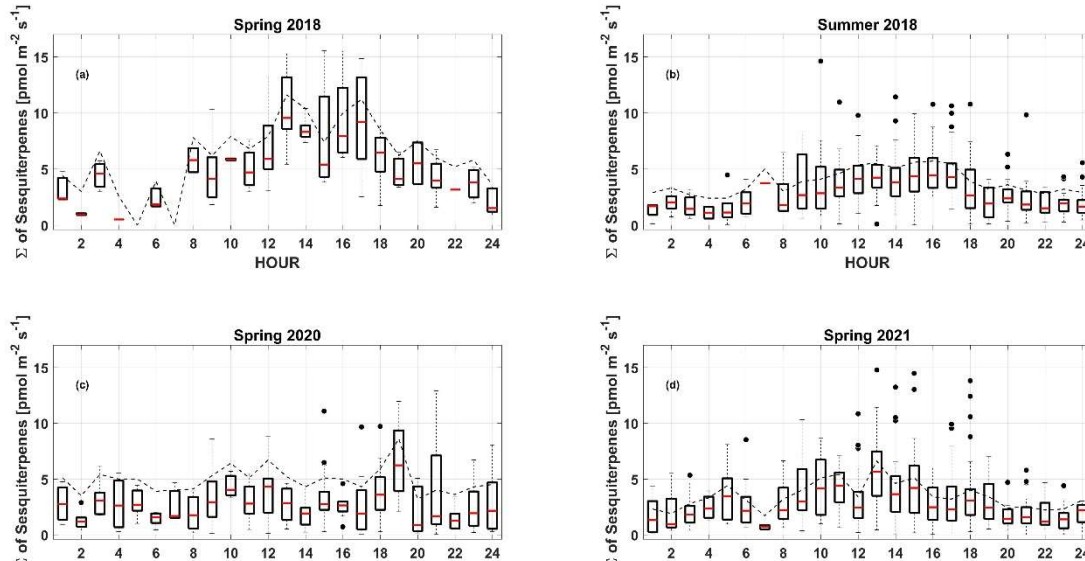

**Figure 6: Daily sum of sesquiterpenes fluxes during spring 2018 (a), summer 2018 (b), spring 2020 (c) and spring 2021 (d). The dotted line represents the average hourly emissions taking into account a 40 % loss caused by reactions with ozone (considering the range of 35-45 % based on Sakulynontvittaya et al., (2008)).**

Due to low urban fluxes during the spring period, disentangling sesquiterpene fluxes into anthropogenic and biotic components is even more difficult than for isoprene and monoterpenes. As for the monoterpenes, the emissions of the sesquiterpenes show

a less pronounced trend in the daily flux pattern (Fig. 6). The measured average sesquiterpenes flux was on the order of 7 pmol $m^{-2} s^{-1}$ in the spring of 2018, 0.004 nmol $m^{-2} s^{-1}$ in the summer of 2018, 1 pmol $m^{-2} s^{-1}$ in 2020 and 2 pmol $m^{-2} s^{-1}$ in 2021. The measured sesquiterpene fluxes may be underestimated by 35-45 % due to losses caused by reacting with ozone (Kaser et al. 2022). The overall ratio between isoprene to monoterpene flux in spring (summer) was 0.97 ([nmol/$m^2$/s] / [nmol/$m^2$/s]) (7.90 ([nmol/$m^2$/s] / [nmol/$m^2$/s])). The ratio between sesquiterpene and monoterpene flux after correcting for a

35-45 % chemical loss was 0.05 ([nmol/$m^2$/s] / [nmol/$m^2$/s]) (spring) and 0.05 ([nmol/$m^2$/s] / [nmol/$m^2$/s]) (summer).

The daily course of emissions shows that the nighttime emissions of sesquiterpenes are low during the weekend in spring compared to the weekdays (data not shown). The analysis of the weekday to weekend variation shows that it was not significant (p-value 0.65) for sesquiterpenes. It suggests a mostly biogenic influence on urban sesquiterpene emissions in Innsbruck.

Compared to the other BVOCs analysed in this study, sesquiterpenes emissions in the urban environment remain low. This is

in agreement with other studies in which low fluxes of sesquiterpenes have been reported in an urban context (Sakulynontvittaya et al., 2008; Kaser et al., 2022).

The correlation between sesquiterpenes and monoterpenes exhibited an $R^2$ of 0.6, with a typical flux ratio of 0.05 ([nmol/$m^2$/s] / [nmol/$m^2$/s]). The correlation coefficient between sesquiterpenes and benzene was comparable to monoterpenes. We interpret these findings to indicate that the correlation between monoterpenes and benzene can only partially be used as a proxy to

estimate the anthropogenic component of monoterpene emissions, because a significant fraction of biogenic terpene emissions seems to interfere with this analysis. We have independently estimated the anthropogenic and biogenic components of these terpene fluxes, and find that they can contribute up to 43 % in spring, but are generally small during the peak of the growing season.

## 3.4 Methanol

In the case of methanol, the sources of emissions can be many and varied. In fact, in addition to having both biological and anthropogenic origins, anthropogenic sources are quite diverse and therefore a quantitative breakdown has not been attempted here.

From a general point of view, during the daytime (8 UTC to 16 UTC), methanol emissions have increased over the years in 485 spring. In 2018 the average daily emission flux was 1.92 nmol m$^{-2}$ s$^{-1}$, while in 2020 it increased to 3.04 nmol m$^{-2}$ s$^{-1}$, and in 2021 it reached 3.41 nmol m$^{-2}$ s$^{-1}$. In this case, the difference in methanol emissions does not seem to be explained by the decrease in temperature during the spring seasons (Table 1). In order to demonstrate this, the temperatures and PAR during the spring and summer periods of 2018 were compared (Fig. 7).

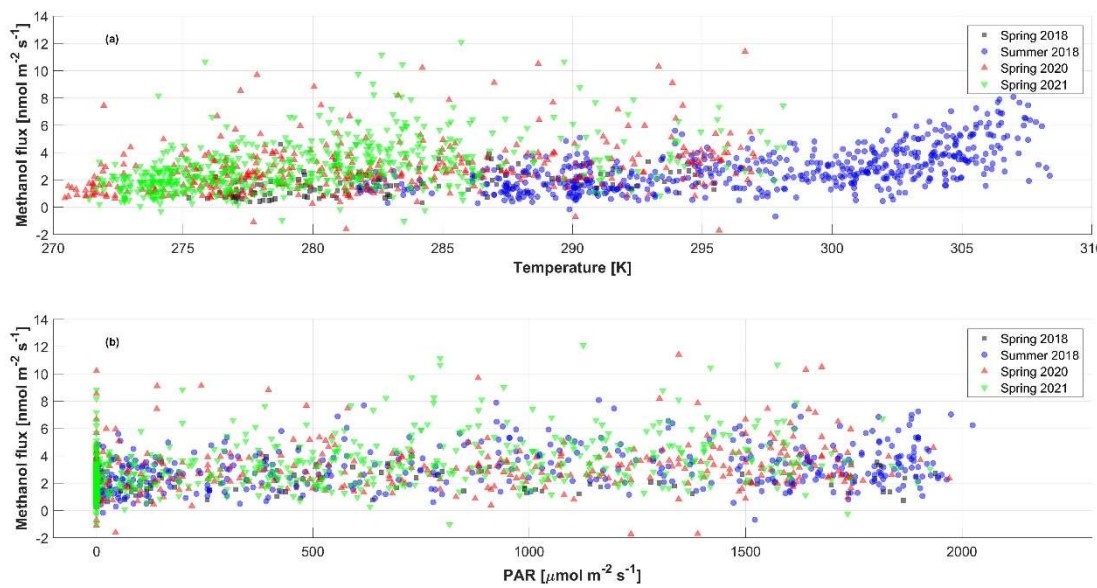


**Figure 7: Methanol fluxes during the four campaigns, spring 2018 (black), summer 2018 (blue), spring 2020 (red), spring 2021 (green), as function of temperature (a) and light (b).**

During the spring period (Fig. 7), it was not possible to detect any meaningful relationship between methanol fluxes and temperature and radiation. In the summer, there is a slight temperature dependence of methanol fluxes above 300 K (Fig. 7a).

The correlation between PAR and methanol emissions is poor for all datasets (Fig. 7b). No temperature and PAR dependence during spring qualitatively suggests little biogenic influence (Wohlfahrt et al., 2015). The diurnal variation of the biogenic methanol flux is strongly driven by stomatal conductance (Niinemets and Reichstein, 2003; Harley et al., 2007), which, provided plants are not water-stressed, usually follows PAR and thus air temperature (Hörtnagl et al., 2011). On the other hand, a difference was observed between the spring and the summer season by analysing the weekday to weekend effect (Table

2). It can be seen that the spring emissions during the weekdays are higher than the emissions during the weekends and the holidays (Fig. 3). The increase in emissions during the weekdays is therefore likely to be the result of an increase in the contribution from anthropogenic sources. The spring-time weekend to weekday effect is comparable to isoprene and monoterpenes, but in summer the weekend to weekday effect was smaller, suggesting a higher contribution of biogenic emissions. During the lockdown period and spring 2021, methanol emission fluxes during weekdays and weekend show

significant changes compared to the 2018 spring campaigns. This was interpreted as methanol emissions are closely related to traffic emissions or emissions related to human presence on the roads.

## 4 Summary

Urban fluxes of common biogenic VOC classes, such as isoprene, monoterpenes, sesquiterpenes and methanol along with anthropogenic tracer VOCs were measured during four campaigns during the late winter-early spring period in 2018, 2020 and

2021 and summer 2018. In-depth analyses of the fluxes made it possible to assess how emissions from anthropogenic urban sources affects emissions of these species that are normally predominantly associated with biogenic sources.

During the seasonal period of spring, the anthropogenic contribution to the total isoprene flux was greater than the contribution from biogenic sources. Approximately 64 % of isoprene emissions are attributable to anthropogenic sources, such as emissions from the transport sector. This finding is consistent with other studies on anthropogenic sources from traffic (Borbon et al.,

2001, 2002; Hellèn et al., 2012; Liu et al., 2024, Khan et al., 2018) and provides an estimate applicable to other urban areas with similar site characteristics. Summer time isoprene emissions are dominated by biogenic emissions with an upper limit of <15 % for the anthropogenic fraction. Isoprene emission fluxes were about an order of magnitude larger in summer than in spring.

We also found evidence for an anthropogenic contribution to monoterpene fluxes in Innsbruck, which is likely related to the

use of personal care products such as cosmetics and personal hygiene products. From the weekend-weekday effect we find that in spring 43 % of monoterpenes can potentially be related to anthropogenic sources. The anthropogenic contribution decreased to less than 20 % in summer.

For the compounds investigated in this study, we detect a dependence of the emissions on the activities of the population of the city of Innsbruck- Emission changes during holiday periods (Sundays and national holidays) and weekdays were observed

for isoprene, monoterpenes, sesquiterpenes, benzene, toluene, methanol and D5 siloxanes for spring. In the 2020 and 2021 campaigns, the differences between weekdays and weekend were more pronounced. During the 2020 COVID lockdown

additional qualitative evidence was inferred by comparing with emission fluxes of siloxanes (in this case D5), which are linked to the use of personal care products (Horii and Kannan, 2008; Wang et al., 2009). During the lockdown period, the state of Tirol imposed one of the strictest mobility restrictions in Europe, resulting in large decreases of all branches of mobility (cars, bikes, pedestrians). This led to an increase in the ratio of weekday to weekend emissions, especially for benzene (4.9) and toluene (3.4). This confirms that during the weekdays the circulation of cars was linked to the working activities of the population, whereas at weekends people were forced to stay at home, as circulation was only possible for urgent needs.

Springtime flux data allow to quantify the effect of terpene emissions from anthropogenic sources, that was suggested by a number of previous studies (Oz et al., 2015; Nazzaro et al., 2013; Cheng et al., 2018; Zhang et al., 2020; Gzatzelis et al., 2021a; Panopoulou et al., 2020, 2021; Kornbausch et al., 2022). Summer time data show that the anthropogenic components of isoprene, monoterpenes and sesquiterpenes play a minor role. For isoprene this is particularity relevant since most of the isoprene flux observed in Innsbruck occurs during the growing season.

Urban methanol emissions are difficult to trace to a particular source due to a combination of possible anthropogenic and biogenic sources. The analysis of weekends and weekdays does not show significant differences, but only an increasing trend over the years. This increase during the late winter/early spring season does not seem to be affected by any difference in PAR or by any decrease in average temperatures, as is the case in the summer season where we have an increase in temperature. We do not observe a significant trend of NDVI values, (proxy for biogenic sources), which are similar over the periods analysed. During the summer period, the results show that increasing temperatures lead to an increase in methanol emissions, which points towards evaporative emissions. The weekend-weekday effect suggests significant anthropogenic contributions to methanol emissions. There still is a need for a deeper understanding of the different sources of methanol in this context.

The vegetation cover in Innsbruck is low, on the order of 19 %. As a consequence, flux data show evidence of anthropogenic VCP emissions contributing to the abundance of urban monoterpenes. In spring this is comparable to the fraction of anthropogenic isoprene, which is released by vehicular emissions. Despite the low vegetation cover, however, isoprene, monoterpene, and sesquiterpene fluxes, which are considered the most important BVOCs, are mostly associated with biogenic emissions during the peak of the growing season.

This characterisation of the different contributions to total terpene emissions on a seasonal basis is fundamental to understanding the possible measures that can be taken to mitigate the effects of these compounds in the atmosphere, which are known to affect air quality. Further studies are also needed during the autumn and winter seasons, which are currently less studied. This study shows that there are no significant variations in total monoterpene fluxes between spring and summer. What does vary is the different contribution from anthropogenic and biogenic sources. When analyzing VCP emissions in context of urban environments we caution that short-term campaign-based observations might over- or underestimate their significance depending on local and seasonal circumstances. Our observations show that significant seasonal variations in absolute emission fluxes as well as the relative apportionment between anthropogenic and biogenic contributions exist.

## Code and data availability

The eddy covariance flux code used to analyse fluxes was published by Striednig et al. (2020) and can be accessed via the following link: https://git.uibk.ac.at/acinn/apc/innflux. VOC and basic micrometeorological data used in this paper are also provided via Zenodo under doi: 10.5281/zenodo.10943991. NDVI data were obtained via https://urs.earthdata.nasa.gov . Additional meteorological data can be obtained from  https://data.hub.geosphere.at/ .

## Author contributions

AP, TK, GW, and MG drafted the manuscript, which was edited by all co-authors. Eddy covariance observations and associated data-analysis and interpretation were performed by AP, MG, TK, CL and MS.

## Competing interest

At least one of the (co-)authors is a member of the editorial board of *Atmospheric Chemistry and Physics*.

## Disclaimer

Publisher's note: Copernicus Publications remains neutral with regard to jurisdictional claims in published maps and institutional affiliations.

## Acknowledgements and financial support

This work was supported by the Austrian National Science fund (FWF) through grant P33701 and P30600.

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
