# Peer review of "Deciphering anthropogenic and biogenic contributions to selected NMVOC emissions in an urban area"

_EGUsphere, 2024_

## Referee Comment (RC1)

Peron et al. provide an analysis of VOC flux measurements conducted over multiple years during the spring and summer seasons in Innsbruck, Austria. The authors focus on quantifying the fluxes of isoprene, monoterpenes, and sesquiterpenes in order to assess the potential contribution of anthropogenic sources, such as motor vehicle emissions and VCPs. Some measurement periods include the COVID lockdown, which provides a unique opportunity to evaluate fluxes in the absence of traffic and pedestrians around the sampling site. The authors evaluate nighttime data, weekday / weekend differences, and seasonal differences to infer the contribution of anthropogenic and biogenic sources to these VOCs.

In general, I find the authors approach to apportioning monoterpenes, methanol, and sesquiterpenes fluxes very informative and an advance in research aimed at quantifying the effects of VCPs and other anthropogenic sources on VOCs that are traditionally biogenic. The weekday / weekend and COVID analyses show changes to the flux that provides bounds on the impact of anthropogenic sources.

My biggest concern relates to the isoprene apportionment. As outlined below in my comments, I am not yet convinced that PTR-ToF-MS measurements are reliable in determining anthropogenic isoprene fluxes. It is not clear to me if the authors correct for anthropogenic interferences to m/z 69.070, which have recently been shown to significantly degrade PTR-ToF-MS measurements of isoprene at night. Consequently, I hope to see more analysis and/or measurement validation that confirms the presence of nighttime isoprene.

**Major Comments:**

The authors use nighttime data to determine the anthropogenic contributions to isoprene emissions. The authors note the importance of fragmentation on the isoprene mass (m/z 69.070) and provide a brief discussion about the potential measurement inferences. As written, it is not clear if the authors corrected the isoprene mass for interferences, or if this discussion is intended to provide error bounds in the flux estimates. Ultimately, I believe a correction is needed, not a discussion of errors, since it is likely that anthropogenic interferences to isoprene overwhelm the signal associated with anthropogenic isoprene and impact the authors' conclusions about weekday/weekend effects, seasonality, and mobile source contributions to isoprene.

My concern is due to the significant contribution of fragmentation to m/z 69.070 previously observed in urban nighttime data. Coggon et al. (2023) showed that interferences to the isoprene mass in urban areas are highest at night and largely associated with the fragmentation from VOCs of anthropogenic origin – i.e., C5 – C9 aldehydes emitted from cooking and possibly other human activities. Coggon et al. show that, at night, the isoprene interference in four urban areas (Los Angeles, Las Vegas, Detroit, New York City) amounts to > 90% of the signal at m/z 69.070. Coggon et al. were able to determine nighttime isoprene mixing ratios after correcting the data, and this was only confirmed by comparison with GC-MS measurements.

Since the authors are using nighttime data to determine the anthropogenic component, I would like to see more discussion / analysis to confirm that indeed a nighttime interference has been

removed. Currently, the authors quote a 30% measurement uncertainty. Is this over the entire day, or is this specific for nighttime measurements? Is there a strong correlation between m/z 69 and aldehyde water-loss products (e.g., m/z 111 + 125) at night that would be indicative of an anthropogenic interference? I believe that the authors need to remove the signals associated with these masses, as higher carbon aldehydes are more indicative of anthropogenic interferences than C5 compounds (e.g. m/z 87), which were attributed by Fall et al. (2001) to be associated with biogenic emissions of alcohols and aldehydes. Even with such an analysis, I would be wary of the isoprene signals at night unless there are GC-MS measurements available to cross-validate the PTR measurements.

**Other Comments:**

Lines 38 – 45: I think this section could be significantly shortened. While it's important to note that BVOCs globally important, I prefer the authors' focus on the impact of BVOCs on urban air quality.

Lines 46 – 48: It would be great if the authors could expand a bit more on the literature that has quantified the impact of BVOCs on urban OH reactivity, ozone formation, and SOA potential. This would be a good place to quantify the isoprene impacts on SOA in China from Wu et al. (2020). Other research could be highlighted as well. For example, Gu et al. examines the role of BVOCs on air quality in Los Angeles and how changing emissions due to urban greening programs might further degrade urban air quality. Pfannerstill et al. conducted aircraft flux measurements in LA and showed that over half of the OH reactivity and SOA formation potential was linked to terpenoids (some biogenic, some anthropogenic). The authors also show that biogenic inventories used in LA significantly underestimated the flux of isoprene – this highlights the importance of flux measurements, such as those presented here by the authors.

Lines 68 – 78: It would be worth noting Borbon et al. (2023) here as well. They show the ubiquity of urban monoterpenes and suggest that monoterpenes emitted in developing countries may have a traffic source. This also highlights the need for identifying mixed source contributions in urban areas.

Section 2.1: It would be useful to see a map of the location, wind direction, and footprint for measurement period, similar to what is shown by Kaser et al. (2022).

Lines 122 – 133: Here, it is not clear how the interferences were treated in the data (see main comment)

Lines 324 – 333: The mass attributed to GLV is potentially impacted by ketones used in VCPs (methyl isobutyl ketone and cyclohexanone, McDonald et al. 2018). It would be worth noting this here.

Section 3.3: There is a lot of great information in this section showing the effects of weekday/weekend, seasonality, and effects of the COVID lockdown period on monoterpene

fluxes. At times, I had trouble keeping all of the points in order. I would find it helpful if this section were broken down a bit more into sub-sections (3.3.1, 3.3.2, etc) that focus on the weekday / weekend effect then the COVID lockdown. For example, at line 390, there could be a Section 3.3.2 that marks the discussion of the lockdown. It would be also helpful to separate the sesquiterpenes with their own sub-section.

Line 382-383: This sentence should be revised. Gkatzelis et al. used monoterpene / benzene ratios as a proxy to evaluate VCP / traffic ratios. The authors attribute monoterpene emissions to VCPs (personal care and cleaning products) rather than traffic emissions.

**References**

Borbon, A., Dominutti, P., Panopoulou, A., Gros, V., Sauvage, S., Farhat, M., et al. (2023). Ubiquity of anthropogenic terpenoids in cities worldwide: Emission ratios, emission quantification and implications for urban atmospheric chemistry. *Journal of Geophysical Research: Atmospheres*, 128, e2022JD037566. https://doi.org/10.1029/2022JD037566

Gu, S., Guenther, A., Faiola, C. Effects of anthropogenic and biogenic volatile organic compounds on Los Angeles Air Quality. *Environmental Science & Technology* **2021** *55* (18), 12191-12201 DOI: 10.1021/acs.est.1c01481

Eva Y. Pfannerstill, Caleb Arata, Qindan Zhu, Benjamin C. Schulze, Roy Woods, Colin Harkins, Rebecca H. Schwantes, Brian C. McDonald, John H. Seinfeld, Anthony Bucholtz, Ronald C. Cohen, and Allen H. Goldstein *Environmental Science & Technology* **2023** *57* (41), 15533-15545, DOI: 10.1021/acs.est.3c03162

---

## Author Comment (AC1)

We thank reviewer 1 for the constructive comments. Below is our point by point reply to specific comments.

Peron et al. provide an analysis of VOC flux measurements conducted over multiple years during the spring and summer seasons in Innsbruck, Austria. The authors focus on quantifying the fluxes of isoprene, monoterpenes, and sesquiterpenes in order to assess the potential contribution of anthropogenic sources, such as motor vehicle emissions and VCPs. Some measurement periods include the COVID lockdown, which provides a unique opportunity to evaluate fluxes in the absence of traffic and pedestrians around the sampling site. The authors evaluate nighttime data, weekday / weekend differences, and seasonal differences to infer the contribution of anthropogenic and biogenic sources to these VOCs. In general, I find the authors approach to apportioning monoterpenes, methanol, and sesquiterpenes fluxes very informative and an advance in research aimed at quantifying the effects of VCPs and other anthropogenic sources on VOCs that are traditionally biogenic. The weekday / weekend and COVID analyses show changes to the flux that provides bounds on the impact of anthropogenic sources.

My biggest concern relates to the isoprene apportionment. As outlined below in my comments, I am not yet convinced that PTR-ToF-MS measurements are reliable in determining anthropogenic isoprene fluxes. It is not clear to me if the authors correct for anthropogenic interferences to m/z 69.070, which have recently been shown to significantly degrade PTR-ToF-MS measurements of isoprene at night. Consequently, I hope to see more analysis and/or measurement validation that confirms the presence of nighttime isoprene.

**Reply:** We thank the referee for pointing out this issue. In the manuscript we have attempted to assess the magnitude of this interference. As suggested by the reviewer we have significantly expanded the analysis as outlined by Coggon et al. and provide a better constraint on the isoprene interference. Overall our main conclusions do not change as outlined below.

**Major Comments:**

The authors use nighttime data to determine the anthropogenic contributions to isoprene emissions. The authors note the importance of fragmentation on the isoprene mass (m/z 69.070) and provide a brief discussion about the potential measurement inferences. As written, it is not clear if the authors corrected the isoprene mass for interferences, or if this discussion is intended to provide error bounds in the flux estimates. Ultimately, I believe a correction is needed, not a discussion of errors, since it is likely that anthropogenic interferences to isoprene overwhelm the signal associated with anthropogenic isoprene and impact the authors' conclusions about weekday/weekend effects, seasonality, and mobile source contributions to isoprene. My concern is due to the significant contribution of fragmentation to m/z 69.070 previously observed in urban nighttime data. Coggon et al. (2023) showed that interferences to the isoprene mass in urban areas are highest at night and largely associated with the fragmentation from VOCs of anthropogenic origin – i.e., C5 – C9 aldehydes emitted from cooking and possibly other human activities. Coggon et al. show that, at night, the isoprene interference in four urban areas (Los Angeles, Las Vegas, Detroit, New York City) amounts to > 90% of the signal at m/z 69.070. Coggon et al. were able to determine nighttime isoprene mixing ratios after correcting the data, and this was only confirmed by

comparison with GC-MS measurements. Since the authors are using nighttime data to determine the anthropogenic component, I would like to see more discussion / analysis to confirm that indeed a nighttime interference has been removed. Currently, the authors quote a 30% measurement uncertainty. Is this over the entire day, or is this specific for nighttime measurements? Is there a strong correlation between m/z 69 and aldehyde water-loss products (e.g., m/z 111 + 125) at night that would be indicative of an anthropogenic interference? I believe that the authors need to remove the signals associated with these masses, as higher carbon aldehydes are more indicative of anthropogenic interferences than C5 compounds (e.g. m/z 87), which were attributed by Fall et al. (2001) to be associated with biogenic emissions of alcohols and aldehydes. Even with such an analysis, I would be wary of the isoprene signals at night unless there are GC-MS measurements available to cross-validate the PTR measurements.

**Reply:** We thank the reviewer for pointing out this issue. We have investigated the interference patterns on m/z 69+ due to fragmentation of higher aldehydes ((e.g., m/z 111 + 125) in more detail. In our case, an analysis was performed both on concentrations, as in Coggon et al. and on fluxes, given the fact that a lot of the analysis in this paper is based on fluxes. In our case, the campaigns were analysed individually, reporting the measured m/z 69 values and those corrected by Coggon et al. With the exception of the spring 2018 campaign, where the measured and corrected values differ slightly from each other, the difference is generally found to be much smaller than that found in Coggon et al. For the other three campaigns, the difference between the two sets of values is negligible (see plots further below). In our study, therefore, we can assume that in the case of m/z 69 there was a comparably small interference with other masses. Three major reasons explain this finding. First, we typically operate the PTR-TOF-MS at lower E/N (ie. 108 Td) as compared to VOCUS type models, that are typically run at 140-160 Td, thus likely yielding a higher degree of fragmentation. Yesildagli et al. for example report almost complete fragmentation of nonanal (https://doi.org/10.1016/j.jhazmat.2023.131368) onto m/z 69. Under regular PTR conditions a fragmentation of 30-40% is observed at most. Second, on the reported aircraft campaigns also the PTR-TOF-MS was also run at higher E/N (e.g. 120 Td) than in this study. Third the average abundance of higher aldehydes as compared to the cited studies seems quite a bit lower. As suggested by the reviewer we provide uncertainty bounds for m/z69 for the assignment of isoprene due to these interferences.

**Other Comments:**

**Lines 38 – 45:** I think this section could be significantly shortened. While it's important to note that BVOCs globally important, I prefer the authors' focus on the impact of BVOCs on urban air quality.

**Reply:** We have rephrased this section, to make it more clear, but still believe mentioning the global significance of BVOCs remains an important aspect worth mentioning in a couple of sentences.

**Lines 46 – 48:** It would be great if the authors could expand a bit more on the literature that has quantified the impact of BVOCs on urban OH reactivity, ozone formation, and SOA potential. This would be a good place to quantify the isoprene impacts on SOA in China from Wu et al. (2020). Other research could be highlighted as well. For example, Gu et al. examines the role of BVOCs on air quality in Los Angeles and how changing emissions due to urban greening programs might further degrade urban air quality. Pfannerstill et al. conducted aircraft flux measurements in LA and showed that over half of the OH reactivity and SOA formation potential was linked to terpenoids (some biogenic, some anthropogenic). The authors also show that biogenic inventories used in LA significantly underestimated the flux of

isoprene – this highlights the importance of flux measurements, such as those presented here by the authors.

**Reply:** Thank you for this comment. We have expanded discussion on this issue and included the references of the relevant literature pointed out by the reviewer. (45-55).

**Lines 68 – 78**: It would be worth noting Borbon et al. (2023) here as well. They show the ubiquity of urban monoterpenes and suggest that monoterpenes emitted in developing countries may have a traffic source. This also highlights the need for identifying mixed source contributions in urban areas.

**Reply:** Thank you for this comment. Observations on the paper cited have been added to the text (80-85).

**Section 2.1:** It would be useful to see a map of the location, wind direction, and footprint for measurement period, similar to what is shown by Kaser et al. (2022).

**Reply:** Thank you for this comment. We added the footprint analysis in the supplement.

**Lines 122 – 133:** Here, it is not clear how the interferences were treated in the data (see main comment)

**Lines 324 – 333:** The mass attributed to GLV is potentially impacted by ketones used in VCPs (methyl isobutyl ketone and cyclohexanone, McDonald et al. 2018). It would be worth noting this here.

Reply: we added the reference and a sentence to that issue. Since the correlation is poor ($R^2$=0.35), any potential influence of isobutyl ketone or cyclohxanone is likely small. However, please take into consideration the fact that the two masses (Methyl isobutyl ketone ($[C_6H_{12}O]H^+$, m/z 101.0966) and cyclohexanone ($[C_6H_{10}O]H^+$, m/z 99.081) reported by the reviewer are not distinguishable from other GLVs through the measurements with the PTR.

**Section 3.3:** There is a lot of great information in this section showing the effects of weekday/weekend, seasonality, and effects of the COVID lockdown period on monoterpene fluxes. At times, I had trouble keeping all of the points in order. I would find it helpful if this section were broken down a bit more into sub-sections (3.3.1, 3.3.2, etc) that focus on the weekday / weekend effect then the COVID lockdown. For example, at line 390, there could be a Section 3.3.2 that marks the discussion of the lockdown. It would be also helpful to separate the sesquiterpenes with their own sub-section.

**Reply:** Thank you for your suggestion. We have divided this part into monoterpenes and sesquiterpenes with subsections according to the type of analysis (weekday/weekend and lockdown).

**Line 382-383:** This sentence should be revised. Gkatzelis et al. used monoterpene / benzene ratios as a proxy to evaluate VCP / traffic ratios. The authors attribute monoterpene emissions to VCPs (personal care and cleaning products) rather than traffic emissions.

**Reply:** Yes, we confirm. We have therefore changed both the sentence in these lines and at 415.

**References**

Borbon, A., Dominutti, P., Panopoulou, A., Gros, V., Sauvage, S., Farhat, M., et al. (2023). Ubiquity of anthropogenic terpenoids in cities worldwide: Emission ratios, emission quantification and implications for urban atmospheric chemistry. Journal of Geophysical Research: Atmospheres, 128, e2022JD037566. https://doi.org/10.1029/2022JD037566

Gu, S., Guenther, A., Faiola, C. Effects of anthropogenic and biogenic volatile organic compounds on Los Angeles Air Quality. Environmental Science & Technology 2021 55 (18), 12191-12201 DOI: 10.1021/acs.est.1c01481

Eva Y. Pfannerstill, Caleb Arata, Qindan Zhu, Benjamin C. Schulze, Roy Woods, Colin Harkins, Rebecca H. Schwantes, Brian C. McDonald, John H. Seinfeld, Anthony Bucholtz, Ronald C. Cohen, and Allen H. Goldstein Environmental Science & Technology 2023 57 (41), 15533-15545, DOI: 10.1021/acs.est.3c03162

**Interference m/z 69**

In our analysis we have taken into account the interference at m/z 69 caused by m/z 87, m/z 111, m/z 125, m/z 129 and m/z 143 detected in all the campaigns analysed in this study. The analysis was performed on both fluxes and concentrations for comparison with Coggon et al. 2023 (https://egusphere.copernicus.org/preprints/2023/egusphere-2023-1497/egusphere-2023-1497.pdf).

[Figure]

Figure 1: fluxes in [nmol m-2 s-1] of m/z 69 compared with those of m/z 87, m/z 111, m/z 125, m/z 129 and m/z 143 for all campaigns analysed without applying any filtering to the data.

Accordingly, we have applied the formula 1 proposed by Coggon et al, 2023:

m/z 69 $_{Corrected}$ = $S_{69}$ − $S_{111 + 125}$ · $f_{69/(111+125)}$ (Eq. 1)

$S_{69}$ is the signal measured at m/z 69, $S_{111+125}$ is the signal of the isoprene interference (sum of m/z111 + m/z 125), and $f_{69/(111+125)}$ is the interference ratio determined at night.

For the determination of the interference ratio at night, only night data from 20 to 3 UTC were extracted. This was based on Coggon et al, 2023 to exclude biogenic sources of m/z 69.

[Figure]

Figure 2: Fluxes in [ nmol m$^{-2}$ s$^{-1}$] measured by m/z 69 in blue and fluxes corrected by applying the correction given by Equation 1 of Coggon et al. 2023 in red.

[Figure]

Figure 3: Measured fluxes (in blue) and corrected fluxes (in red) during the day.

Looking instead at concentrations:

[Figure]

Figure 4: Concentrations in [ppb] of m/z 69 compared to those of m/z 87, m/z 111, m/z 125, m/z 129 and m/z 143 for all campaigns analysed without applying any filter to the data.

[Figure]

Figure 5: concentrations in [ppb] m/z 69 in blue and corrected concentrations using the correction given by Equation 1 of Coggon et al. 2023 in red. This is for each campaign.

[Figure]

Figure 6: measured flows (in blue) and corrected flows (in red) during the day.

It can be seen that the values reported in Figure 6 follow a similar trend to that reported in Figure 4 in Coggon et al., 2023 for the Las Vegas site.

Coggon et al, 2023 report an $f_{69/(111+125)}$ of 3 for the city of Las Vegas. In our study the average $f_{69/(111+125)}$ is 0.35 for the spring period and 0.21 for the summer period. The nocturnal interference ratio found in the city of Innsbruck is therefore lower for all periods analysed, compared to that found by Coggon et al. 2023 in the city of Las Vegas.

This allowed us to consider the possible interference found by the PTR-MS at m/z 69 as small, and thus to consider the values measured at this m/z as those of isoprene emissions. Especially in the 2020 campaign, during the lockdown. As suggested by the reviewer, we now provide realistic bounds for assigning isoprene to m/z 69, based on upper limit interferences from m/z87, m/z111, and m/z 125.

---

## Author Comment (AC2)

We thank reviewer 2 for the constructive comments. Below is our point by point reply to specific comments.

Peron et al. used eddy covariance measurements of terpenoids and other VOCs to estimate their contributions from anthropogenic and biogenic sources in the city of Innsbruck, Austria.

The paper is an interesting contribution to the emerging subject of volatile chemical product emissions in urban areas, where it is especially challenging to find out how much of the terpenoids is anthropogenic. As such, it is especially important since there are, to date, few analyses of VCP emissions in European cities. The manuscript is well written, and I recommend its publication in ACP after the following comments have been addressed:

**General:** The paper mentions a lot of correlation analyses that were done for the data analysis but shows none of the plots. I think it would be beneficial to add supplementary material that makes at least the most important correlation analyses accessible and verifiable for the reader.

**Reply:** Thank you for these comments. We have added the most important plots that are useful to understand the correlation analysis.

**For the discussion:** It would be interesting to compare the derived anthropogenic emissions to emission factors per person used in the literature (e.g. Coggon et al. 2021 use an estimate monoterpene emission factor per person), since the population in the footprint of the authors' station should be available.

Coggon, M. M., Gkatzelis, G. I., McDonald, B. C., Gilman, J. B., Schwantes, R. H., Abuhassan, N., Aikin, K. C., Arend, M. F., Berkoff, T. A., Brown, S. S., Campos, T. L., Dickerson, R. R., Gronoff, G., Hurley, J. F., Isaacman-VanWertz, G., Koss, A. R., Li, M., McKeen, S. A., Moshary, F., Peischl, J., Pospisilova, V., Ren, X., Wilson, A., Wu, Y., Trainer, M., and Warneke, C.: Volatile chemical product emissions enhance ozone and modulate urban chemistry, PNAS, 118, 1–9, https://doi.org/10.1073/pnas.2026653118, 2021.

**Reply:** This is a very valuable suggestion. However it is not trivial, because Coggon et al. did not measure emissions directly. Rather they report concentration enhancements and concentrations. In their figure 2 they show that monoterpene concentrations go up to about 70 ppt in center of NYC where the population density peaks at ~20k/km2. In their figure 1 they show a monoterpene/benzene correlation plot with population density, increasing from about 0.2 to 1.2. We can infer a slope of about 5.2e-5 [ppb/ppb]/[people/km$^2$] from their work. For the spring time data, we obtain a monoterpene/benzene flux ratio of about 0.35. The average population density was previously estimated to lie between 8800 during the day and 7000 during night (see Ward et al., 2022: https://doi.org/10.5194/acp-22-6559-2022). This leaves us with population normalized monoterpene to benzene ratio of 3.9e-5 to 5.0e-5 [(nmol/m$^2$/s)/(nmol/m$^2$/s)]/[people/km$^2$].

**l. 39ff:** Are these % by mass or molar? This makes an important difference. The first sentence (50% of terpene emissions are isoprene, which means that the other 50% must be monoterpenes+sesquiterpenes) contradicts the second one (15% + 0.5% do not add up to 50%).

**Reply:** We have clarified this sentence. According to Guenther et al. 2012 (https://doi.org/10.5194/gmd-5-1471-2012), "the global total BVOC flux of about 1000 Tg is speciated into ∼ 50 % isoprene, and ∼ 15 % monoterpenes), with the rest (i.e. 35%) comprised by other BVOC such as oxygenated VOCs. It is similar to results from previous models although emissions for specific locations and days may be very different. A dozen compounds have annual global emissions exceeding 1 % of the global total and together they comprise ∼ 80 % of the total flux. Isoprene has been the most studied of these compounds. In contrast, there are relatively few studies of CO, ethene, propene and ethanol emissions that can be used to parameterize or evaluate their emissions. It is clear that isoprene is the globally dominant BVOC and should continue to be the focus of BVOC emissions research although other compounds with a greater capacity for producing aerosol matter (e.g., monoterpenes and sesquiterpenes) or impacting the upper atmosphere (e.g., methyl halides) may have equally or even more important roles in specific earth system processes. In addition, compounds such as MBO may have a small annual global emission but are important for specific regions and seasons. MEGAN2.1 does not include a generic "other VOC" category as was the case for previous models (e.g., Guenther et al., 1995). Instead MEGAN includes only specific compounds so that they can be used in atmospheric chemistry schemes. Many of the compounds listed in Table 1 have relatively low estimated emission rates and so contribute little to the total estimated BVOC flux. The relatively large uncertainties in these rates do not rule out the possibility of higher contributions and these compounds should be considered in future emission measurement studies.

**l. 44:** Unclear: do you refer to total emission strength from plants or of global methanol emissions?

**Reply:** This refers to biogenic emissions, i.e. those emitted by vegetation.

**l. 55:** It is true that simple models based (just) on future temperatures predict an increase in BVOC terpenoid emissions, but I think the authors should acknowledge that there are other factors that play a role (e.g. increased CO2 inhibits isoprene fluxes), so the situation is way more complicated, as shown in this review: Holopainen, J. K., Virjamo, V., Ghimire, R. P., Blande, J. D., Julkunen-Tiitto, R., and Kivimäenpää, M.: Climate Change Effects on Secondary Compounds of Forest Trees in the Northern Hemisphere, Frontiers in plant science, 9, 1445, https://doi.org/10.3389/fpls.2018.01445, 2018.

As a result, it is impossible to predict how BVOC will change globally with climate change. This is e.g. discussed in the latest IPCC report (Szopa, S., v. Naik, Adhikary, B., Artaxo, P., Berntsen, T., Collins, W. D., Fuzzi, S., Gallardo, L., Kiendler-Scharr, A., Klimont, Z., Liao, H., Unger, N., and Zanis, P.: Short-Lived Climate Forcers, in: Climate Change 2021: The Physical Science Basis. Contribution of Working Group I to the Sixth Assessment Report of the Intergovernmental Panel on Climate Change, edited by: Masson-Delmotte, V., Zhai, P., Pirani, A., Connors, S. L., Péan, C., Berger, S., Caud, N., Chen, Y., Goldfarb, L., Gomis, M. I., Huang, M., Leitzell, K., Lonnoy, E., Matthews, J., Maycock, T. K., Waterfield, T., Yelekçi, O., Yu, R., and Zhou, B., Cambridge University Press, Cambridge, United Kingdom and New York, NY, USA, 817–922, https://doi.org/10.1017/9781009157896.008, 2021.)

**Reply:** Thank you for this comment. Observations on the papers cited have been added to the text (65-70).

**l. 122 ff:** Regarding interferences on m/z 69.07: Firstly, it is commendable that the authors acknowledge the presence of interferences on the protonated isoprene mass. However: Did the authors do their described correlation analysis also just using nighttime data? Given the low isoprene fluxes at night in

general, the interference from cooking aldehydes may be more important during the night than on average. I would like to see the respective correlation plots in the Supplement.

An interference of 30% from m/z 87.08 plus an interference of <30% from higher aldehydes seems not insignificant if anthropogenic isoprene is 64% in the spring – given the potential interference discussed by the authors, it could be that a significant fraction of that is actually from an aldehyde interference and thus likely from cooking.

So, at least the percentages given e.g. in the abstract should include uncertainties, and I wish the authors would also consider in the discussion that they may partly see the influence of cooking emissions contributing to anthropogenic m/z 69.07. It would be even better if they could correct their isoprene fluxes for the interference(s).

**l. 402:** Maybe I am wrong, but wasn't the $R^2$ of 0.48 in Gkatzelis et al. related to population density, not benzene?

**Reply:** Yes, we confirm. We have therefore changed both the sentence in these lines and at lines 395-400

**l. 478:** The Borbon et al. studies were done more than 20 years ago. Are there any newer studies to confirm that cars with modern catalysts still emit isoprene?

**Reply:** The catalyst for gasoline cars was introduced in the 80ies and 90ies, and by the time when Borbon conducted measurements already quite well established across Europe, but we have also added newer references on this subject. (e.g. Khan et al., 2018: https://www.mdpi.com/2073-4433/9/10/387 , Liu et al., 2024). Khan et al. 2018 for example found similar contribution of urban traffic in London.

**l. 482:** also fragranced cleaning products?

**Reply:** Yes, we include the cleaning products in the personal hygiene products (e.g. laundry detergents that also contain fragrances related to monoterpenes).

**l. 516:** I am missing an explanation/hypothesis as to why the sum of monoterpene fluxes stays the same between spring and summer, but the anthropogenic fraction changes. Do people use less fragranced products in the summer? (Seems a little unlikely to me.)

**Reply:** This is a good point. Our observed fluxes are similarly high during the seasons, which might come as a surprise, since biogenic emissions are expected to be higher in summer. We note that the anthropogenic part was derived from the weekend to weekday effect. We also saw a much poorer correlation between benzene and monoterpene fluxes in summer, both suggesting a less significant influence of anthropogenic emissions during summer (e.g. ~20%). The campaign happened to take place during a significant heat wave in 2018, when people preferentially stayed in cooler environments (not in the unpleasant climate of the urban heat island on streets). A possible explanation of this effect could therefore be that fewer people were out on the streets during the day in the summer of 2018. While we do not have explicit data on people mobility outside, it could provide an explanation of this effect. Another important factor that needs to be considered and is related to this issue is that fewer students are on the University campus due to the summer break. The University of Innsbruck and the Medical University of Innsbruck together host about 30.000 students. An additional explanation could be that hot temperatures let fragrances on clothes evaporate more quickly, or simply that people wear fewer clothes during

summer. We would suspect that a significant amount of fragrances are also linked to washed clothing with detergents being evaporated.

These consideration are related to:

Traffic during the spring and summer, where we see a significant effect of the analysis of the weekend weekday effect (line 393 circa).

Correlation with D5 during the period of the lockdown, where we measure emissions of D5 during the weekdays (when the people where aloud to go outside to buy groceries and partially work) and not during weekend as people were not allowed to leave their homes, except for special reasons. In addition, we reported similar results for this particular period for monoterpene fluxes.

The correlation between D5 siloxane and benzene fluxes observed here, only exhibited an $R^2$ lower than <0.13, thus was very poor. In contrast, Gkatzelis et al. (2021a) observed a very high correlation with $R^2$ of ~0.8. However, D5 siloxane fluxes observed here are generally very low, especially in spring. From the spring analysis, the weekend to weekday variation of monoterpenes was similar to isoprene (e.g. a ratio of 1.9 for isoprene vs a ratio of 1.6 for monoterpenes). This suggests a similar partitioning between anthropogenic and biogenic emission components in spring. As a best estimate from various constraints, we argue that 50-67 % of isoprene and monoterpenes emissions during spring can be associated with anthropogenic activity, but less than 20 % during summer. In this context it is noteworthy to mention that isoprene fluxes are about a factor of 10 higher during summer (Fig. 4), while monoterpenes fluxes vary much less throughout the seasons and the magnitude is comparable during the seasons (Fig. 5).

In this study, a good correlation between monoterpenes and D5 reported by Gkatzelis et al. (2021a) was not present during the nighttime and during the lockdown period (data not shown). This is an indication that the anthropogenic component of monoterpene fluxes was lower during the lockdown period, and consequently was more associated with vegetation due to the strict mobility restrictions in Austria (Lamprecht et al., 2021).

**Data availability**: I do not think it is enough to make the data available on request. The authors should upload the final flux data to a publicly accessible server with a DOI. E.g., Zenodo makes this very easy.

**Reply:** The flux data for isoprene, monoterpenes, sesquiterpenes, benzene, toluene, methanol and siloxane used in this paper were provided via the Zenodo platform.

**Interference m/z 69**

In our analysis we have taken into account the interference at m/z 69 caused by m/z 87, m/z 111, m/z 125, m/z 129 and m/z 143 detected in all the campaigns analysed in this study. The analysis was performed on both fluxes and concentrations for comparison with Coggon et al. 2023 (https://egusphere.copernicus.org/preprints/2023/egusphere-2023-1497/egusphere-2023-1497.pdf).

[Figure]

Figure 1: fluxes in [nmol m-2 s-1] of m/z 69 compared with those of m/z 87, m/z 111, m/z 125, m/z 129 and m/z 143 for all campaigns analysed without applying any filtering to the data.

Accordingly, we have applied the formula 1 proposed by Coggon et al, 2023:

m/z 69 $_{Corrected}$ = $S_{69}$ − $S_{111+125}$ · $f_{69/(111+125)}$ (Eq. 1)

$S_{69}$ is the signal measured at m/z 69, $S_{111+125}$ is the signal of the isoprene interference (sum of m/z111 + m/z 125), and $f_{69/(111+125)}$ is the interference ratio determined at night.

For the determination of the interference ratio at night, only night data from 20 to 3 UTC were extracted. This was based on Coggon et al, 2023 to exclude biogenic sources of m/z 69.

[Figure]

Figure 2: Fluxes in [ nmol m⁻² s⁻¹] measured by m/z 69 in blue and fluxes corrected by applying the correction given by Equation 1 of Coggon et al. 2023 in red.

[Figure]

Figure 3: Measured fluxes (in blue) and corrected fluxes (in red) during the day.

Looking instead at concentrations:

[Figure]

Figure 4: Concentrations in [ppb] of m/z 69 compared to those of m/z 87, m/z 111, m/z 125, m/z 129 and m/z 143 for all campaigns analysed without applying any filter to the data.

[Figure]

Figure 5: concentrations in [ppb] m/z 69 in blue and corrected concentrations using the correction given by Equation 1 of Coggon et al. 2023 in red. This is for each campaign.

[Figure]

Figure 6: measured flows (in blue) and corrected flows (in red) during the day.

It can be seen that the values reported in Figure 6 follow a similar trend to that reported in Figure 4 in Coggon et al., 2023 for the Las Vegas site.

Coggon et al, 2023 report an $f_{69/(111+125)}$ of 3 for the city of Las Vegas. In our study the average $f_{69/(111+125)}$ is 0.35 for the spring period and 0.21 for the summer period. The nocturnal interference ratio found in the city of Innsbruck is therefore lower for all periods analysed, compared to that found by Coggon et al. 2023 in the city of Las Vegas.

This allowed us to consider the possible interference found by the PTR-MS at m/z 69 as small, and thus to consider the values measured at this m/z as those of isoprene emissions. Especially in the 2020 campaign, during the lockdown. As suggested by the reviewer, we now provide realistic bounds for assigning isoprene to m/z 69, based on upper limit interferences from m/z87, m/z111, and m/z 125.